# Dataset Distillation via Factorization

**Songhua Liu    Kai Wang    Xingyi Yang    Jingwen Ye    Xinchao Wang**
National University of Singapore
{songhua.liu,e0823044,xyang}@u.nus.edu, {jingweny,xinchao}@nus.edu.sg

## Abstract

In this paper, we study dataset distillation (DD), from a novel perspective and introduce a *dataset factorization* approach, termed *HaBa*, which is a plug-and-play strategy portable to any existing DD baseline. Unlike conventional DD approaches that aim to produce distilled and representative samples, *HaBa* explores decomposing a dataset into two components: data *Ha*llucination networks and *Ba*ses, where the latter is fed into the former to reconstruct image samples. The flexible combinations between bases and hallucination networks, therefore, equip the distilled data with exponential informativeness gain, which largely increase the representation capability of distilled datasets. To furthermore increase the data efficiency of compression results, we further introduce a pair of adversarial contrastive constraints on the resultant hallucination networks and bases, which increase the diversity of generated images and inject more discriminant information into the factorization. Extensive comparisons and experiments demonstrate that our method can yield significant improvement on downstream classification tasks compared with previous state of the arts, while reducing the total number of compressed parameters by up to 65%. Moreover, distilled datasets by our approach also achieve ~10% higher accuracy than baseline methods in cross-architecture generalization. Our code is available here.

## 1 Introduction

The success of deep models on a variety of vision tasks, such as image classification [23, 9, 34], object detection [33, 32], and semantic segmentation [39, 49, 25], is largely attributed to the huge amount of data used for training and various pre-trained models [50]. However, the sheer amount of data introduces significant obstacles for storage, transmission, and data pre-processing. Besides, publishing raw data inevitably brings about privacy or copyright issue in practice [40, 8]. To alleviate these problems, Wang *et al*. [47] pioneer the research of dataset distillation (DD), to distill a large dataset into a synthetic one with only a limited number of samples, so that the training efforts with the distilled dataset for downstream models on the original dataset can be largely reduced, which facilitates a series of applications like continual learning [37, 36, 48, 27] and black-box optimization [6]. Due the significant practical value of DD, many endeavours have been made on this area [55, 53, 54, 46, 21, 5, 56] to design novel supervision signals to train the synthetic datasets and to further improve their performances.

Nevertheless, there is a potential drawback in conventional settings of DD: it largely treats each synthetic sample independently and ignores the inter coherence and relationship between different instances. As such, the information embraced by each sample, despite distilled, is by nature limited. Using the synthetic samples for training downstream models, therefore, inevitably leads to the loss of dataset information. Moreover, the few distilled samples are incompatible with the enormous number of parameters in a deep model and may yield the risk of overfitting.

To verify these potential issues, we conduct a pre-experiment on CIFAR10 dataset with 10 synthetic images per class, using MTT [5], the current SOTA solution on DD, as the baseline. In addition

36th Conference on Neural Information Processing Systems (NeurIPS 2022).

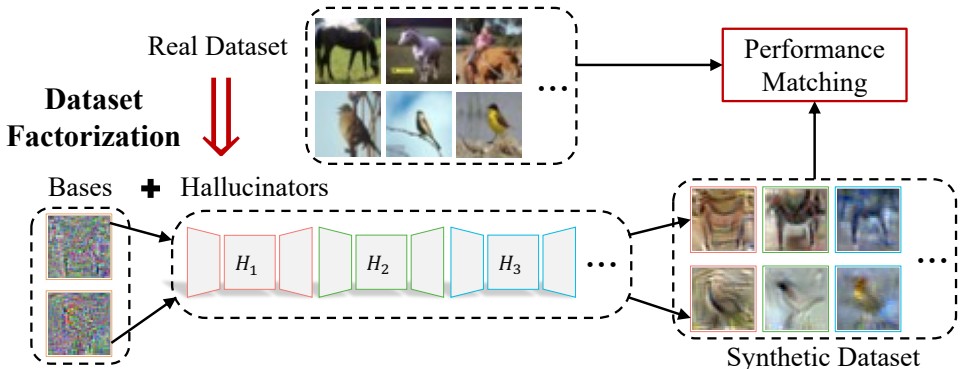

Figure 1: Intuition of our hallucinator-basis factorization for dataset distillation.

to the baseline setting, we also incorporate all the checkpoint synthetic datasets after each 100 DD iterations in the convergent stage to train the downstream model. Since the synthetic images are fine-tuned during this stage, multiple checkpoints can be viewed as related but different, which may somehow increase the diversity. As a result, it yields overall lower test loss and hence better final results in downstream training, as shown in the blue and green curves in Fig. 2, which indicates that current DD solutions can be potentially improved by leveraging some sample-wise relationships to diversify the distilled data. Nevertheless, simply involving more data samples may also increase the memory overhead. This fact motivates us to ask: *is it possible to encode some shared relationships in a dataset implicitly, instead of storing samples directly, to avoid such additional storage costs?*

We show in this paper that, it can indeed be made possible through reformulating the DD task as a *factorization* problem. As shown in Fig. 1, we propose a novel perspective dubbed *HaBa*, to factorize a dataset into two compositions: data *Ha*llucination networks and *Ba*ses. A data hallucination network, or hallucinator, can take any basis as input and output the corresponding hallucinated image. Supervised by the training objective of DD, a set of hallucinators can synthesize multiple samples from a common basis and are optimized to extract effective relationships among different samples in original datasets explicitly. In this way, information of $|\mathcal{H}| \times |\mathcal{B}|$ images can be included for a factorization result with $|\mathcal{H}|$ hallucinators and $|\mathcal{B}|$ bases via arbitrary pair-wise combination, which improves the data efficiency of traditional DD exponentially. As shown in the yellow curve in Fig. 2, with the same budget on the storage, our strategy achieves better test performance compared with the MTT baseline.

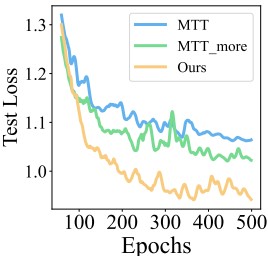

Figure 2: Visualization of test loss using synthetic datasets generated by MTT, MTT with multiple checkpoints, and ours.

To further increase the informativeness of factorized results, we introduce a pair of adversarial contrastive constraints to promote sample-wise diversity. The goal of HaBa is to minimize the correlation among images composed of different hallucinators but a common basis, while an adversary tries to maximize it. Such an adversarial scheme, in turn, enforces the hallucinators to produce diversified images and increases the amount of useful information.

Notably, HaBa is a versatile strategy that can be built upon existing DD baselines, since it is compatible with any training objective for measuring the similarity between downstream performances as shown in Fig. 1, We conduct extensive experiments to demonstrate the advantages of the proposed method over baseline ones. In all benchmarks and comparisons, HaBa produces significant and consistent improvement on training downstream models, while reducing the total number of compressed parameters by up to 65%. Furthermore, it demonstrates strong cross-architecture generalization ability with accuracy improvement higher than 10%. Our contributions are summarized as follows:

- We study dataset factorization, a novel perspective to explore dataset distillation, and propose a novel approach termed HaBa for hallucinator-basis factorization.

- We present a pair of adversarial contrastive objectives to further increase the data diversity and information capability.

- HaBa is a plug-and-play scheme compatible with all existing training objectives of DD and can yield significant and consistent improvement over the state of the arts.

## 2   Related Works

The goal of dataset distillation (DD) is to optimize a smaller synthetic dataset such that it is capable to take place of original one for training downstream tasks, which is different from coreset selection [1, 7, 13, 38, 44], another branch for dataset compression, directly selecting samples from raw datasets. In this section, we provide a detailed review of previous methods in DD.

Motivated from knowledge distillation [15, 12, 52, 51] aiming at model compression, Wang *et al.* [47] introduce the concept of dataset distillation for dataset compression. The idea is to optimize the synthetic images so that they can minimize loss functions of downstream tasks, where a bilevel optimization algorithm [11] is involved. Following this routine, several works further consider learnable labels beyond samples [3, 42]. Subsequently, Zhao *et al.* [55] and several following approaches [53, 24] consider matching gradients of a downstream model produced by synthetic samples and real images, which improve the performance significantly. Most recently, Cazenavette *et al.* [5] argue that single-iteration gradient matching may lead to inferior performance due to error accumulation across multiple steps and thereby propose to match long-range training dynamics of an expert trained on the original dataset. As an alternative method to profile training effects produced by different sets, Nguyen *et al.* [29, 30] also introduce the kernel ridge-regression approach based on the Neural Tangent Kernel (NTK) in infinitely wide convolutional networks [17].

Apart from matching training effects, there are also methods matching data distributions between original and synthetic datasets. For instance, Zhao *et al.* [54] propose a simple but effective Maximum Mean Discrepancy (MMD) constraint for DD, which does not involve the training of downstream models and enjoys superior training efficiency. Wang *et al.* [46] propose CAFE, explicitly attempting to align the synthetic and real distributions in the feature space of a downstream network.

Above mentioned methods are dedicated to exploring suitable training objectives and pipelines for DD. However, there are few works concerning improving the data efficiency for distilled samples. Although Zhao *et al.* [53] propose differentiable siamese augmentation (DSA) to enrich the training data, the augmentation operations used, *e.g.*, crop, flip, scale, and rotation, cannot encode any information about the target datasets. In this paper, we study the task in a factorization perspective, to factorize a dataset into two different compositions: data hallucination networks and bases. Both parts carry important knowledge of the raw dataset. For downstream training, hallucinators and bases can perform arbitrary pair-wise combination, *i.e.*, sending any basis to any hallucinator, to create a training sample. The idea of factorization can improve the diversity of distilled training datasets significantly, without introducing additional costs for storage. It is also a versatile strategy compatible with all aforementioned DD methods, which will be demonstrated in the experiment part.

**Concurrent Works on Efficient Distilled Dataset Parameterization:** As a concurrent work, Kim *et al.* [21] propose IDC for efficient synthetic data parameterization. It reveals that only storing down-sample version of synthetic images and conducting bilinear upsampling in downstream training would not hurt the performance much. Thus, given the same budget of storage, it can store $4\times$ number of $2\times$ down-sample synthetic images compared with the baseline. Both IDC and HaBa in this paper are dedicated to improving the data efficiency of synthetic parameters. Interestingly, according to the definition of our hallucinator-basis factorization, IDC can in fact be treated as a special case of HaBa, where the hallucinator is a parameter-free upsampling function and each basis has a smaller spatial size. Nevertheless, the main focuses for IDC and HaBa are different and they are in fact two orthogonal techniques, which can readily join force to enhance the baseline performance, as discussed in Sec. 4.2.

## 3   Methods

In this section, we elaborate our proposed method *HaBa* for dataset distillation (DD). Assume that there is an original dataset $\mathcal{T} = \{(x_i, y_i)\}_{i=1}^{|\mathcal{T}|}$ with $|\mathcal{T}|$ pairs of a training sample $x_i$ and the corresponding label $y_i$. DD targets a synthetic dataset $\mathcal{S} = \{(\hat{x}_i, \hat{y}_i)\}_{i=1}^{|\mathcal{S}|}$ with $|\mathcal{S}| \ll |\mathcal{T}|$ and expects that a model trained on $\mathcal{S}$ can have similar performance than that trained on $\mathcal{T}$.

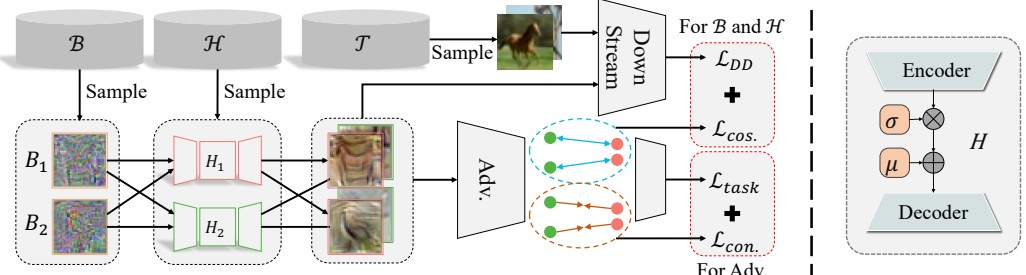

Figure 3: Left: Overall pipeline of the proposed hallucinator-basis factorization. $\mathcal{B}$, $\mathcal{H}$, and $\mathcal{T}$ denote sets of bases, hallucinators, and original data respectively. *Adv.* denotes an adversary model. We adopt batch size 2 here for clarity; Right: Architecture of a hallucinator in detail.

Traditional DD methods treat each synthetic sample independently and ignore the inner relationship between different samples within a dataset, which results in poor data/information efficiency. Focusing on such drawback, we study DD from a novel perspective and redefine it as a hallucinator-basis factorization problem:

$$S = \{H_{\theta_j}\}_{j=1}^{|\mathcal{H}|} \cup \{(\hat{x}_i, \hat{y}_i)\}_{i=1}^{|\mathcal{B}|}, \tag{1}$$

where there are $|\mathcal{H}|$ hallucination networks and $|\mathcal{B}|$ bases. The $j$-th hallucinator is parameterized by $\theta_i$ and we denote it by $H_{\theta_i}$ for $1 \leq j \leq |\mathcal{H}|$. For downstream training, a training data pair $(\tilde{x}_{ij}, \tilde{y}_{ij})$ is created online via sending the $i$-th basis, with any $1 \leq i \leq |\mathcal{B}|$, to the $j$-th hallucinator, with any $1 \leq j \leq |\mathcal{H}|$, *i.e.*, $\tilde{x}_{ij} = H_{\theta_j}(\hat{x}_i)$. In this paper, the label $\tilde{y}_{ij}$ is simply taken as $\hat{y}_i$.

An overview of our method is shown in Fig. 3(Left). To go deeper into the technical details, we first start with the introduction of our basis and data hallucination network in Sec. 3.1. Then, we propose an adversarial contrastive constraint to increase data diversity in Sec. 3.2. Finally, we present the whole training pipeline of the hallucinator-basis factorization for DD in Sec. 3.3.

## 3.1   Basis and Hallucinator

**Basis:** Typically, for an image classification dataset $\mathcal{T} = \{(x_i, y_i)\}_{i=1}^{|\mathcal{T}|}$, $x_i \in \mathbb{R}^{h \times w \times c}$ and $y_i \in \{0, 1, ..., C-1\}$ for each $1 \leq i \leq |\mathcal{T}|$, where each $x_i$ is a $c$-channel image with a resolution of $h \times w$, and $C$ is the total number of classes. In previous DD methods, the format/shape of synthetic data pairs $(\hat{x}, \hat{y})$ has to be held the same as that of real data, so as to make sure the consistency between input and output formats in the training and test time for downstream models. By contrast, since hallucinator networks are capable of spatial-wise and channel-wise transformation, the shape of each $\hat{x}_i$, $1 \leq i \leq |\mathcal{B}|$, denoted as $h' \times w' \times c'$, is not necessarily the same as that of original samples and thus more flexible. And for a classification problem, we do not modify its label space in this paper for simplicity and maintain the categorical format.

**Hallucinator:** Given a basis $\hat{x} \in \mathbb{R}^{h' \times w' \times c'}$, a data hallucination network, aims to create a new image $\tilde{x} \in \mathbb{R}^{h \times w \times c}$ based on $\hat{x}$, which can be viewed as a conditional image generation problem. Inspired by image style transfer [19, 16, 18, 26], a typical conditional image generation problem, we devise an encoder-transformation-decoder based architecture for hallucinators, as shown in Fig. 3(Right). Specifically, the encoder, denoted as $enc$, is composed of CNN blocks, which non-linearly maps an input $\hat{x}$ to a feature space $\mathbb{R}^{h'' \times w'' \times c''}$. Then, an affine transformation with scale $\sigma$ and shift $\mu$ is conducted on the derived feature, where $\sigma$ and $\mu$ are treated as network parameters in this paper. At last, the decoder $dec$ under a symmetric CNN architecture with $enc$ projects the transformed feature back to the image space. Formally, this process can be written as:

$$\hat{f} = enc(\hat{x}), \quad \tilde{f} = \sigma \times \hat{f} + \mu, \quad \tilde{x} = dec(\tilde{f}), \tag{2}$$

where the multiplication is element-wise operation. There are $|\mathcal{H}|$ hallucinators in the whole factorization pipeline and each would be trained to implicitly encode some sample-wise relations by its network parameters.

## 3.2   Adversarial Contrastive Constraint

Ideally, the knowledge encoded by different hallucinators should be as different/orthogonal as possible to get the most benefits for each individual. To instantiate such regularization, let's consider two

composed images $\tilde{x}_{ij}$ and $\tilde{x}_{ik}$ from two different hallucinators $H_{\theta_j}$ and $H_{\theta_k}$ but a common basis $\hat{x}_i$. The divergence between $\tilde{x}_{ij}$ and $\tilde{x}_{ik}$ is expected to be large. To measure the divergence, a feature extractor is required to map an input image to a feature space, and how to train such a feature extractor to find an appropriate feature space is of great importance.

In this paper, we formalize the training of hallucinators and the feature extractor as a min-max game in a self-consistent manner, where the feature extractor desires to minimize the divergence between $\tilde{x}_{ij}$ and $\tilde{x}_{ik}$ while hallucinators, as well as bases, are optimized to maximize it so that the two players can reinforce each other. In specific, the feature extractor, denoted as $F$ and parameterized by $\psi$, is typically a CNN structure for the downstream task and we adopt features at the last hidden layer before the output layer, denoted as $F_{-1}(\tilde{x}_{ij})$ and $F_{-1}(\tilde{x}_{ik})$. $F$ is optimized to maximize the correlation between the two feature vectors, which can be quantified by the metric of mutual information (MI). Inspired by the lower bound of MI [45], the objective to minimize the divergence for $F$ is given by the following contrastive form:

$$\mathcal{L}_{con.} = -\frac{1}{|\mathcal{H}|^2}\frac{1}{|\mathcal{B}|}\sum_{\substack{1 \leq j,k \leq |\mathcal{H}|, \\ j \neq k}}\sum_{i=1}^{|\mathcal{B}|}\log\frac{\exp\{F_{-1}^\top(\tilde{x}_{ij})F_{-1}(\tilde{x}_{ik})/\tau\}}{\sum_{u=1}^{|\mathcal{B}|}\exp\{F_{-1}^\top(\tilde{x}_{ij})F_{-1}(\tilde{x}_{uk})/\tau\}}, \tag{3}$$

where $\tau$ is a scalar temperature coefficient. For the classification problem, we can alternatively adopt the supervised form of the contrastive loss $\mathcal{L}_{con.}$, where $\tilde{x}_{uk}$ with the same class label as $\tilde{x}_{ij}$ are also taken into consideration as positive samples in Eq. 3. The supervised contrastive loss can benefit to increase the correlation of samples from the same class [20] for a more reasonable feature representation.

In addition, the feature space is expected to reflect the task-specific property for a meaningful representation. Thus, we also incorporate the task loss $\mathcal{L}_{task}$, *e.g.*, cross-entropy loss in classification tasks, over the synthetic dataset as a supervision signal for $F$. In this way, the overall training objective for $F$ is defined as:

$$\min_{\psi} \mathcal{L}_F = \lambda_{con.}\mathcal{L}_{con.} + \lambda_{task}\mathcal{L}_{task}, \tag{4}$$

where $\lambda_{con.}$ and $\lambda_{task}$ are hyper-parameters controlling the weight for each term.

$F$ acts as an adversary to minimize the divergence between $\tilde{x}_{ij}$ and $\tilde{x}_{ik}$, while the synthetic dataset is expected to maximize it to increase data diversity. To this ends, the similarity between $F_{-1}(\tilde{x}_{ij})$ and $F_{-1}(\tilde{x}_{ik})$ becomes one loss term for hallucinator-basis factorization. In this paper, we adopt the cosine-similarity and the objective $\mathcal{L}_{cos.}$ is given by:

$$\mathcal{L}_{cos.} = \frac{1}{|\mathcal{H}|^2}\frac{1}{|\mathcal{B}|}\sum_{\substack{1 \leq j,k \leq |\mathcal{H}|, \\ j \neq k}}\sum_{i=1}^{|\mathcal{B}|}\frac{F_{-1}^\top(\tilde{x}_{ij})F_{-1}(\tilde{x}_{ik})}{\|F_{-1}(\tilde{x}_{ij})\|_2\|F_{-1}(\tilde{x}_{ik})\|_2}. \tag{5}$$

During training, the feature extractor and the factorized components are updated alternately to play this min-max game.

### 3.3 Factorization Training Pipeline

Following previous paradigms [55, 54, 5, 46], the synthetic dataset $\mathcal{S}$ is updated in an iterative algorithm. In each iteration, we randomly sample a batch of hallucinators and bases and conduct pair-wise combinations. The composed images are evaluated by the objective of dataset distillation $\mathcal{L}_{DD}$ and the similarity metric in Eq. 5:

$$\min_{\mathcal{S}} \mathcal{L}_\mathcal{S} = \lambda_{DD}\mathcal{L}_{DD} + \lambda_{cos.}\mathcal{L}_{cos.}, \tag{6}$$

where hyper-parameters $\lambda_{DD}$ and $\lambda_{cos.}$ balance the loss.

Notably, the hallucinator-basis factorization is compatible with a variety of configurations of $\mathcal{L}_{DD}$ by previous arts, which makes it a versatile and effective strategy for DD. In this paper, we adopt the trajectories matching loss in Cazenavette *et al*. [5] as $\mathcal{L}_{DD}$ by default thanks to its superior performance. The basic idea is to update a downstream model from a cached checkpoint $\phi_t^*$ at iteration $t$, using the synthetic dataset $\mathcal{S}$ for $N$ times, and using the real dataset $\mathcal{T}$ for $M$ times

respectively. The updated parameters by the two cases, $\hat{\phi}_{t+N}$ and $\phi^*_{t+M}$ are enforced to be consistent:

$$\hat{\phi}_{t+n+1} \leftarrow \hat{\phi}_{t+n} - \alpha \nabla_{\hat{\phi}_{t+n}} \mathcal{L}_{task}(\mathcal{S}), \quad \hat{\phi}_t \leftarrow \phi^*_t, \quad 0 \leq n < N,$$
$$\phi^*_{t+m+1} \leftarrow \phi^*_{t+m} - \beta \nabla_{\phi^*_{t+m}} \mathcal{L}_{task}(\mathcal{T}), \quad 0 \leq m < M, \tag{7}$$
$$\mathcal{L}_{DD} = \frac{\|\hat{\phi}_{t+N} - \phi^*_{t+M}\|_2^2}{\|\phi^*_t - \phi^*_{t+M}\|_2^2},$$

where $\alpha$ and $\beta$ are learning rates with $\mathcal{S}$ and $\mathcal{T}$ respectively. $\alpha$ is learnable in the framework while $\beta$ is a hyper-parameter. In Sec. 4.2, we also experiment with other settings of $\mathcal{L}_{DD}$.

Based on the supervised signals in Eq. 6, the gradients are backward propagated to the composed images and finally to the sampled hallucinators and bases so as to be updated using a decent algorithm such as SGD. Since all the operations are differentiable, the training can be completed end-to-end.

## 4 Experiments

### 4.1 Datasets and Implementing Details

We conduct evaluations of our method on three standard image classification benchmarks: SVHN [28], CIFAR10, and CIFAR100 [22]. There are 60,000 images for real-world digit recognition in SVHN. For CIFAR10 and CIFAR100, there are 50,000 training images in total. The number of classes for the three datasets are 10, 10, and 100 respectively. All the images are under $32 \times 32$ resolution in 3-channel RGB format. Following previous works [5], we use ZCA for image preprocessing with Kornia implementation [35] before all the experiments. Experiments with more datasets, including images in larger spatial scales, can be found in the supplement.

In this paper, for convenience of comparisons with prior works, we maintain the same size with images in original datasets, *i.e.*, $h' = h$, $w' = w$, and $c' = 3$ for bases. We also experiment with other sizes of bases in Sec. 4.3. For hallucinators, the encoder and decoder contain 1 Conv-ReLU blocks. The number of feature channel $c''$ is 3. We use 5 hallucinators by default. The learning rates of hallucinators and bases, $\eta_H$ and $\eta_B$, are the same and for the feature extractor, the learning rate $\eta_F$ is 0.001. Hyper-parameters $\lambda_{con.}$, $\lambda_{task}$, $\lambda_{DD}$, and $\lambda_{cos.}$ are set as 0.1, 1, 1, and 0.1 empirically. Sensitivities of these hyper-parameters are analyzed in Sec. 4.3. The adversary network has the same architecture as that for computing $\mathcal{L}_{DD}$. In experiments on SVHN and CIFAR10, we incorporate all the bases in each iteration, while in experiments on CIFAR100, we adopt a batch size of 300 when the total number of bases is greater than 1,000. We only consider random 2 hallucinators in one iteration for simplicity. The maximal configuration of computational resources is 4 24GB 3090 GPUs. The GPU memory consumption is dependent on that of the baseline method for $\mathcal{L}_{DD}$ and is slightly higher than it due to the computation of $\mathcal{L}_{cos.}$ and $\mathcal{L}_{con.}$. The baseline method for $\mathcal{L}_{DD}$ is MTT [5] if not specified. Other settings related to DD hold the same as the baseline. All the quantitative results are based on the mean and standard deviation over 5 repeated experiments. To make sure fair comparisons, the dataset size in our method is equal to the number of bases $|\mathcal{B}|$ and the hallucinators are treated as parameterized data augmentors working online in downstream training, just as general data augmentations, which means that the dataset size does not increase compared with the baselines.

### 4.2 Comparisons

**Comparisons with State of the Arts:** We compare HaBa with previous state of the arts for DD in standard settings, to synthesize 1, 10, and 50 images per class (IPC) respectively. In our setting, the number of parameters in a hallucinator is is approximately equal to that for 2 synthetic images, while the size of a basis is equal to that of an image. Taking the storage cost of 5 hallucinators into consideration, we set the number of bases per class (BPC) as IPC minus 1 in each IPC configuration when IPC is greater than 1, to make the comparisons as fair as possible. Candidates are coreset based methods including Random [7, 31], Herding [4, 2], K-Center [10, 38], and Forgetting [43], meta learning based methods including DD [47] and LD [3], training matching based methods including DC [55], DSA [53], and MTT [5], and distribution matching based methods including DM [54] and CAFE [46]. The comparisons follow the standard protocol adopting a 3-layer Conv-InstanceNorm-ReLU-AvgPool ConvNet with 128 channels in training and testing.

| | Dataset | SVHN | | | CIFAR10 | | | CIFAR100 | | |
|---|---|---|---|---|---|---|---|---|---|---|
| | IPC | 1 | 10 | 50 | 1 | 10 | 50 | 1 | 10 | 50 |
| | Ratio % | 0.014 | 0.14 | 0.7 | 0.02 | 0.2 | 1 | 0.2 | 2 | 10 |
| Coreset | Random | 14.6±1.6 | 35.1±4.1 | 70.9±0.9 | 14.4±2.0 | 26.0±1.2 | 43.4±1.0 | 4.2±0.3 | 14.6±0.5 | 30.0±0.4 |
| | Herding | 20.9±1.3 | 50.5±3.3 | 72.6±0.8 | 21.5±1.3 | 31.6±0.7 | 40.4±0.6 | 8.4±0.3 | 17.3±0.3 | 33.7±0.5 |
| | K-Center | 21.0±1.5 | 14.0±1.3 | 20.1±1.4 | 21.5±1.3 | 14.7±0.9 | 27.0±1.4 | 8.3±0.3 | 7.1±0.2 | 30.5±0.3 |
| | Forgetting | 12.1±1.7 | 16.8±1.2 | 27.2±1.5 | 13.5±1.2 | 23.3±1.0 | 23.3±1.1 | 4.5±0.3 | 9.8±0.2 | - |
| Distillation | DD[†] [47] | - | - | - | - | 36.8±1.2 | - | - | - | - |
| | LD[†] [3] | - | - | - | 25.7±0.7 | 38.3±0.4 | 42.5±0.4 | 11.5±0.4 | - | - |
| | DC [55] | 31.2±1.4 | 76.1±0.6 | 82.3±0.3 | 28.3±0.5 | 44.9±0.5 | 53.9±0.5 | 12.8±0.3 | 25.2±0.3 | - |
| | DSA [53] | 27.5±1.4 | 79.2±0.5 | 84.4±0.4 | 28.8±0.7 | 52.1±0.5 | 60.6±0.5 | 13.9±0.3 | 32.3±0.3 | 42.8±0.4 |
| | DM [54] | - | - | - | 26.0±0.8 | 48.9±0.6 | 63.0±0.4 | 11.4±0.3 | 29.7±0.3 | 43.6±0.4 |
| | CAFE [46] | 42.6±3.3 | 75.9±0.6 | 81.3±0.3 | 30.3±1.1 | 46.3±0.6 | 55.5±0.6 | 12.9±0.3 | 27.8±0.3 | 37.9±0.3 |
| | CAFE+DSA [46] | 42.9±3.0 | 77.9±0.6 | 82.3±0.4 | 31.6±0.8 | 50.9±0.5 | 62.3±0.4 | 14.0±0.3 | 31.5±0.2 | 42.9±0.2 |
| | MTT [5] | 58.5±1.4 | 70.8±1.8 | 85.7±0.1 | 46.3±0.8 | 65.3±0.7 | 71.6±0.2 | 24.3±0.3 | 39.0±0.1 | 46.1±0.2 |
| Factorization | BPC | 1 | 9 | 49 | 1 | 9 | 49 | 1 | 9 | 49 |
| | Ratio % | 0.028 | 0.14 | 0.7 | 0.04 | 0.2 | 1 | 0.22 | 1.82 | 9.82 |
| | HaBa | **69.8±1.3** | **83.2±0.4** | **88.3±0.1** | **48.3±0.8** | **69.9±0.4** | **74.0±0.2** | **33.4±0.4** | **40.2±0.2** | **47.0±0.2** |
| Whole Dataset | | 95.4±0.1 | | | 84.8±0.1 | | | 56.2±0.3 | | |

Table 1: The performance (test accuracy %) comparison to state-of-the-art methods. LD[†] and DD[†] use AlexNet for CIFAR10, while the rest use ConvNet for training and testing. IPC: Number of Images Per Class; BPC: Number of Bases Per Class; Ratio (%): the ratio of distilled images to whole training set. Underline denotes results by our implementation.

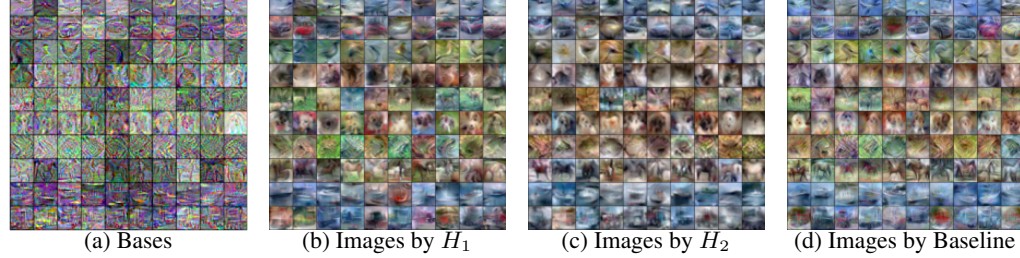

(a) Bases       (b) Images by $H_1$      (c) Images by $H_2$    (d) Images by Baseline

Figure 4: Visualization of factorized results by our HaBa (70.27% test acc.) and baseline MTT (65.92% test acc.). Zoom-in for better comparisons.

The comparison results are shown in Tab. 1 and we can observe that HaBa achieves state-of-the-art performance in all datasets and settings. Especially when the ratio of distilled images to the whole training set is less than 1%, our method can yield significant improvement over all the candidate methods, which demonstrates that the scheme of hallucinator-basis factorization improves the data efficiency for the task of dataset distillation.

**Qualitative Comparisons:** We visualize the factorized results by our method as well as the baseline on CIFAR10 dataset with 10 BPC in Fig. 4. Due to the space limitation, we only provide images generated by 2 hallucinators here. More results can be found in the supplement. As shown in the figure, we can find that bases mainly store some main structures and contour information. Different hallucinators would render a basis with diverse styles and details. Thanks to the dataset factorization scheme, the diversity of distilled images by our method is higher than that by the baseline.

**Building upon Different Baselines:** To reflect the versatility of the insight, we implement HaBa on multiple state-of-the-art training pipelines of DD, including DC, DM, and MTT. We evaluate the performance of synthetic datasets on CIFAR10 and maintain the IPC of baseline methods as BPC plus 1, which makes storage costs for synthetic datasets as close as possible for fairness. As shown in Tab. 2, when training and testing on ConvNet, the strategy of HaBa can make a consistent improvement over all the baselines, which demonstrates that factorization is a general idea to improve the data efficiency in DD.

**Cross-Architecture Performance:** For DD, a satisfactory distilled dataset should have similar training effects to the original one on downstream models with arbitrary architectures. Thus, cross-architecture generalization performance is an important metric for DD. We use the synthetic datasets trained on ConvNet to train models with different structures including ResNet [14], VGG [41], and AlexNet [23]. The results can be found in Tab. 2. Benefiting from the increased data diversity, HaBa can improve the across-architecture accuracy significantly with a performance gain up to 17.57%. The consistent and significant improvement validates the superior ability of our method to capture the informative features and thus original datasets can be replaced by the synthetic ones better.

| | Method | DC [55] | | | DM [54] | | | MTT [5] | | |
|---|---|---|---|---|---|---|---|---|---|---|
| | IPC | 2 | 11 | 51 | 2 | 11 | 51 | 2 | 11 | 51 |
| | BPC | 1 | 10 | 50 | 1 | 10 | 50 | 1 | 10 | 50 |
| ConvNet | Baseline | 31.36±0.16 | 45.29±0.30 | 54.24±0.61 | 34.57±0.52 | 50.35±0.36 | 62.03±0.29 | 50.59±0.95 | 63.90±0.29 | 69.81±0.48 |
| | w. HaBa | 34.11±0.47 | 49.88±0.52 | 58.91±0.23 | 37.32±0.13 | 56.83±0.11 | 64.44±0.40 | 56.76±0.38 | 69.48±0.26 | 73.25±0.21 |
| | Gain | +2.75 | +4.59 | +4.67 | +2.75 | +6.48 | +2.41 | +6.17 | +5.58 | +3.44 |
| ResNet | Baseline | 18.10±0.76 | 18.36±0.36 | 22.14±0.38 | 22.25±1.00 | 40.00±1.49 | 53.40±0.68 | 35.15±0.96 | 45.05±1.46 | 54.47±0.95 |
| | w. HaBa | 24.49±0.55 | 24.27±0.56 | 31.08±0.32 | 31.34±0.72 | 47.57±0.49 | 59.61±0.35 | 47.39±0.71 | 57.97±0.88 | 64.35±0.60 |
| | Gain | +6.39 | +6.11 | +8.94 | +9.09 | +7.57 | +6.21 | +12.24 | +12.92 | +9.88 |
| VGG | Baseline | 28.02±0.26 | 35.88±0.67 | 38.73±0.48 | 22.28±1.03 | 41.64±0.64 | 55.17±0.54 | 38.04±1.19 | 50.49±1.02 | 61.36±0.30 |
| | w. HaBa | 29.42±0.93 | 37.03±0.42 | 41.91±0.55 | 26.93±0.62 | 49.41±0.36 | 67.47±0.43 | 48.26±0.54 | 60.47±0.56 | 67.47±0.43 |
| | Gain | +1.40 | +1.15 | +3.18 | +4.65 | +7.77 | +12.30 | +10.22 | +9.98 | +6.11 |
| AlexNet | Baseline | 20.02±1.31 | 22.42±1.35 | 29.48±0.87 | 20.67±3.64 | 37.04±0.92 | 49.14±0.94 | 26,06±1.01 | 35.95±1.52 | 49.20±1.27 |
| | w. HaBa | 22.24±1.14 | 33.02±0.91 | 33.42±1.39 | 32.14±0.60 | 44.14±0.67 | 53.09±0.89 | 43.63±1.46 | 48.96±3.00 | 60.07±1.37 |
| | Gain | +2.22 | +10.60 | +3.94 | +11.47 | +7.10 | +3.95 | +17.57 | +13.01 | +10.87 |

Table 2: Cross-architecture performance (test accuracy %) comparison to different baseline methods of DD HaBa built upon.

**Comparisons under the Same Number of Final Images:** In the default comparison protocol, we compare our method with the baselines using the same budget of storage, where our method can store information of exponentially more images than the baselines with the same number of parameters. In this part, we also examine the performance of HaBa under the condition that the number of final images, *i.e.*, $|\mathcal{H}| \times |\mathcal{B}|$, is equal to that used by the baseline. Intuitively, given that the objective functions of our method and the baseline are the same exactly, the performance of the baseline can be viewed as an upper bound of ours, since there are significantly less parameters in our method to carry the information of final images in this case. Therefore, we first remove the term $\mathcal{L}_{cos.}$ from the loss function of DD in Eq. 6 to guarantee a consistent optimization objective with the baseline. Then, we compare the performance of HaBa and the baseline using 10, 20, 30, 40, and 50 final images respectively. Here, the number of hallucinators $|\mathcal{H}|$ is 2 and the number of bases is thus half of the number of final images. As shown in the red and green curves in Fig. 7, performance of the baseline can be well approximated by ours with only half of the number of parameters, especially when the number of images is relatively large. Remarkably, with the proposed adversary contrastive constraint, our method can even outperform the baseline consistently, as shown in the blue curve, which further demonstrates the effectiveness of the proposed solution.

**Comparisons with Concurrent Works on Efficient Distilled Dataset Parameterization:** As a concurrent work on efficient distilled dataset parameterization, IDC [21] is proposed to store $4\times$ number of $2\times$ down-sample synthetic images compared with the baseline. The core is to reduce the spatial size for efficient parameterization. For HaBa of this paper, instead, we do not modify the spatial size of bases in the default setting for better qualitative explainablity and more intuitive comparisons with the baselines. In this sense, IDC and HaBa are in fact two orthogonal techniques and they can readily join force to enhance the baseline performance. Here, we try using the technique of IDC and adopting $2\times$ down-sample synthetic images on the baseline MTT, based on which we further consider adding our HaBa and involving 5 hallucinators. As shown in Tab. 3, with the efficient parameterization of IDC, the performance of baseline can be improved. With HaBa in this paper, the performance can even be further improved a lot: 5.14%, 1.29%, and 4.30% in the three settings respectively, which demonstrates that IDC and HaBa work in different ways.

**Applications in Continual Learning:** To further demonstrate the advantage of the proposed method for improving data efficiency, following the setting of DM [54], we conduct experiments on the setting of continual learning on CIFAR-100, with 20 random classes per stage. The average number of parameters per class is $20 \times 32 \times 32 \times 3$. The synthetic datasets are trained with a ConvNet with 3 blocks. We evaluate synthetic datasets by our method and the DM baseline on the same ConvNet architecture and ResNet18. The results in Fig. 5 demonstrate that the proposed method increases the informativeness of synthetic datasets and thus produce significantly better performance, especially in the cross-architecture setting.

### 4.3 Ablation Studies

**Loss Terms:** To validate the effectiveness of the proposed adversarial contrastive constraints, we design ablation studies on the CIFAR10 dataset over three loss terms: $\mathcal{L}_{cos.}$ in Eq. 5, $\mathcal{L}_{con.}$ in Eq. 3, and the task-specific loss $\mathcal{L}_{task}$. Through the results in Tab. 4, we can find that deleting any one of

| # of Param. / Class | 2×32×32×3 | 11×32×32×3 | 51×32×32×3 |
|---|---|---|---|
| Baseline | 49.89±0.95 | 65.92±0.62 | 70.73±0.52 |
| w. IDC | 56.13±0.38 | 70.85±0.43 | 71.01±0.41 |
| w. IDC & HaBa | 61.27±0.34 | 72.14±0.22 | 75.31±0.27 |

Table 3: Comparisons with concurrent work IDC [21] on efficient synthetic parameterization.

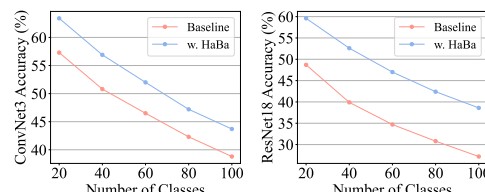

Figure 5: Comparisons on the setting of continual setting. Results on the ConvNet3 (Left) and ResNet18 (Right) architectures are shown.

| BPC | 1 | 10 | 50 |
|---|---|---|---|
| HaBa w/o $\mathcal{L}_{cos.}$ | 54.56±0.61 | 70.16±0.44 | 73.93±0.21 |
| HaBa w/o $\mathcal{L}_{con.}$ | 54.91±0.49 | 70.07±0.48 | 72.50±0.39 |
| HaBa w/o $\mathcal{L}_{task}$ | 54.62±0.42 | 70.07±0.16 | 72.74±0.20 |
| HaBa Full | 55.66±0.29 | 70.27±0.63 | 74.04±0.16 |
| HaBa w $\mathcal{L}_{con.}$ Downstream | 56.78±0.22 | 70.44±0.15 | 75.00±0.52 |

Table 4: Results of ablation study on loss terms in HaBa: $\mathcal{L}_{cos.}$, $\mathcal{L}_{con.}$, and $\mathcal{L}_{task}$.

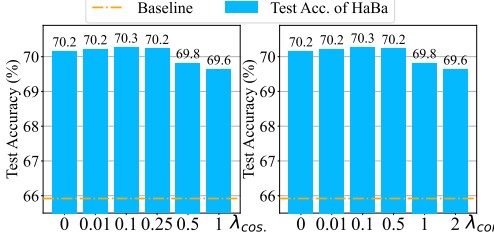

Figure 6: Impacts of different $\lambda_{cos.}$ and $\lambda_{con.}$ on the test accuracy.

them would hurt the performance. We also experiment with involving $\mathcal{L}_{con.}$ for downstream training, to enforce the similarity among images composed of different hallucinators and a common basis. Observed from the last row of Tab. 4, the performance can be further improved, since $\mathcal{L}_{con.}$ helps the representation learning of related samples [20]. Note that we do not use this loss term for downstream training in other experiments for fair and standard comparison.

We examine the sensitivities of hyper-parameters $\lambda_{cos.}$ and $\lambda_{con.}$ used to balance the weights of loss terms $\mathcal{L}_{cos.}$ and $\mathcal{L}_{con.}$ respectively in Fig. 6. The results are evaluated on the CIFAR10 dataset with 10 BPC. We can observe that the overall performance is not sensitive to the selection of these hyper-parameters and our method makes a consistent improvement over the baseline with 11 IPC.

**Class-Independent Hallucinators v.s. Shared Hallucinators:** In the default setting of HaBa, each class maintains a certain number of bases independently and all the classes share the same set of hallucinators. But what about the case that hallucinators are also made class-independent? We study this problem experimentally in Tab. 5. Given the same BPC, class-independent hallucinators can indeed somehow improve the performance when there are fewer synthetic samples, *e.g.*, 1 BPC. However, when BPC is higher, equipping each class with an independent set of hallucinators would not benefit the performance. There are probably two reasons: (1) shared hallucinators across all the classes extract global information of the whole dataset, which encodes more representative and universal knowledge; and (2) the class-independent case would make the number of hallucinators 10 times for the CIFAR10 dataset, which leaves a heavy burden for the optimization process. Thus, as indicated in Tab. 5, a better solution is to make room for more bases using the memory allocated to store class-independent hallucinators initially, which would result in more satisfactory data efficiency.

**Number of Channels Used by Basis:** By default, the shape of a basis is the same as that of a real image, which is generally in RGB 3-channel format. In fact, in Fig. 8(Left), we also verify that it is also possible to use single-channel basis, which can reduce the memory cost by nearly 2/3 without hurting the performance too much. Interestingly, if the memory cost is held the same, we can choose to use 3 times BPC to store single-channel bases, rather than 3-channel ones. This would yield impressive improvement on the test accuracy when BPC is small. Note that for baseline results, IPC is set as the corresponding BPC plus 1.

**Number of Hallucinators:** We study the impact of the number of hallucinators, *i.e.*, $|\mathcal{H}|$, in Fig. 8(Right). We can observe that when BPC is small, including more hallucinators is helpful for the performance. Nevertheless, when BPC is 10 or 50, the performance would not improve with more hallucinators when $|\mathcal{H}| > 10$. One reason is that when $|\mathcal{H}|$ is large, the sampling of hallucinators in each iteration is sparse, which makes the joint optimization of all the hallucinators more difficult.

**Data Augmentation:** The similarity between our hallucinator set and data augmentation lies that both of them can contribute to generating more samples and increasing the diversity. However, the essential difference is that our hallucinators are optimized to encode sample-wise relationships in

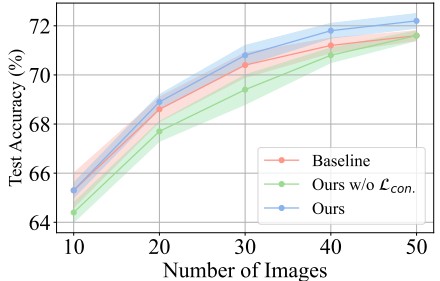

Figure 7: Comparisons with the baseline under the same number of final images.

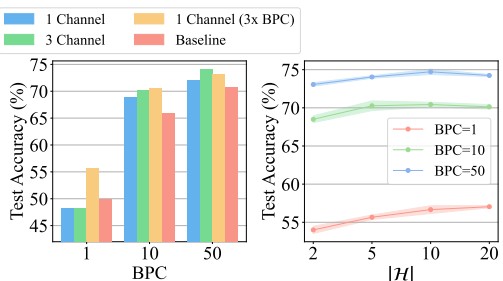

Figure 8: Study on the number of channels used by bases and the number of hallucinators.

| BPC | 1 | 10 | 50 |
|---|---|---|---|
| w/o Share | 55.96±0.51 | 69.00±0.20 | 69.81±0.56 |
| Share | 55.66±0.29 | 70.27±0.63 | 74.04±0.16 |
| Baseline (IPC=BPC) | 45.29±0.86 | 62.77±0.56 | 71.09±0.34 |
| Share (Same Memory) | 70.27±0.63 | 72.17±0.30 | 74.89±0.15 |
| Baseline (Same Memory) | 65.92±0.62 | 68.58±0.49 | 73.55±0.48 |

Table 5: Study on whether all the classes should share the same set of hallucinators.

| | ConvNet | ResNet | VGG | AlexNet |
|---|---|---|---|---|
| w/o aug. | 60.63±0.21 | 43.24±0.83 | 48.02±0.53 | 30.58±1.44 |
| Baseline | 63.90±0.29 | 45.05±1.46 | 50.49±1.02 | 35.95±1.52 |
| w/o aug. | 68.08±0.23 | 56.37±0.11 | 59.04±0.50 | 48.27±3.04 |
| Ours | 69.48±0.26 | 57.97±0.88 | 60.47±0.56 | 48.96±3.00 |

Table 6: Impact of data augmentation.

a dataset, while data augmentation is based on some prior and heuristic knowledge of images. By default, both our method and the baseline adopt the data augmentation strategy DSA [53]. To study the relationship between the two schemes experimentally, we attempt to remove DSA from baseline and our method and report the corresponding results in Tab. 6. The evaluation is on CIFAR10 with 11 IPC for baseline and 10 BPC for ours. Through the results, we can find that (1) our method without data augmentation can also outperform the baseline method with augmentation significantly, which means that the mechanism of HaBa can benefit the dataset distillation task more with the learning of global information of a dataset in hallucinators; and (2) with data augmentation, our performance can be further improved, which indicates that HaBa and DSA work in different manners.

## 5 Conclusions, Limitations, and Future Works

This paper proposes a novel hallucinator-basis factorization method dubbed HaBa for dataset distillation (DD). It uses hallucinators to encode inner relations between different samples in original datasets, which can largely improve the data efficiency of distilled results. To diversify the knowledge captured by different hallucinators, a pair of adversarial contrastive constraints is further introduced. Extensive evaluations and comparisons on multiple benchmark datasets demonstrate that HaBa is capable of significantly improving the performance of downstream models trained on the synthetic dataset, using only 35% cost of memory for storage. Moreover, it is a versatile strategy that is compatible with different configurations of DD frameworks and yields consistent improvement.

Despite the superior performance of the proposed hallucinator-basis factorization (HaBa) scheme, there are also some potential limitations. On the one hand, compared with the baseline method HaBa built upon, the process of online pairwise combination between hallucinators and bases in training increases the cost of time and GPU memory slightly, although light-weight hallucinators are adopted. On the other hand, it may inherited the limitations of baseline methods. For example, when the number of images is large, further increasing the number would produce limited performance gain.

For future works, beyond the training efficiency of HaBa, introducing class-wise relationship may also be a potential research direction. For example, it is probably optimal that one class shares hallucinators with some specific classes but does not share with others. It is also promising to explore more advance factorization for a dataset to further improve the performance.

## Acknowledgement

This research is supported by the National Research Foundation, Singapore under its Medium Sized Centre for Advanced Robotics Technology Innovation (WBS: A-0009428-09-00). Xinchao Wang is the corresponding author.

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
