# OpenReview forum: "Dataset Distillation via Factorization"
_NeurIPS.cc/2022/Conference — NeurIPS 2022 Accept_

### Official Review · Reviewer_jG3P · 2022-06-27

**Rating:** 6
**Confidence:** 2
**Soundness:** 3 good
**Presentation:** 3 good
**Contribution:** 3 good

**Summary:**

This paper proposes a novel algorithm for dataset condensation and introduces a dataset factorization approach - HABA which frame the factorization as a hallucinator-basis problem. This paper also introduces a pair of adversarial contrastive constrains to increase the diversity of generated images and inject more discriminant information into the factorization.

**Questions:**

See Weaknesses.

**Limitations:**

adequately addressed

**Strengths And Weaknesses:**

### Strengths
1. This paper is well written and easy to follow, and the presentation is clear.

2. The overall idea is novel and the experimental results are convincing.

### Weaknesses
I have to be honest that I'm not an expert in this area, but I have a few questions about this paper.

1. Since the author has mentioned that the early dataset compression methods are inspired by the knowledge distillation, I'm wondering if these compressed data can be used for KD methods.

2. Although this paper has performed the experiments across different architectures, however, I'm wondering about the effect of the model size on the final performance. (i.e. the performance gap between the compressed data with the whole data on ResNet18 will be greater or smaller than ResNet-101 ?)

---

> ### Author Response · Authors · 2022-08-02
> **Response to Reviewer jG3P**
>
> We appreciate the reviewer jG3P's efforts on the constructive feedback and are glad that the reviewer finds our work novel and the experimental results convincing. The questions are fully addressed as follows:
>
> 1. **Can the compressed data be used for KD methods?**
>
>    * Thank you for your question. The idea of dataset distillation is indeed inspired by the knowledge distillation. Nevertheless, they are orthogonal techniques which can be applied jointly: to compress both model and data. We conduct an experiment on CIFAR-10 dataset with 10 bases per class:
>
>      |          | w/o KD | w. KD Teacher on Real | w. KD Teacher on Synthetic | w. KD Both Teachers |
>      | :------: | :----: | :-------------------: | :------------------------: | :-----------------: |
>      |   Ours   | 69.48  |         69.70         |           69.82            |        70.03        |
>      | Baseline | 63.90  |         64.43         |           64.57            |        64.95        |
>
>      Here, "Teacher on Real" means using a network trained on the real dataset as the teacher, "Teacher on Synthetic" means using a network trained on the synthetic dataset as the teacher, and "Both Teachers" means including both networks trained on the real dataset and the synthetic dataset respectively as the teacher. The teacher and student networks have 4 and 3 convolution blocks respectively.
>
> 2. **The effect of model size on the final performance.**
>
>    * Thank you for the insightful and interesting question. We conduct an experiment on the CIFAR-10 dataset with 10 bases per class. The model used for dataset distillation is a standard convolutional neural network with 3 convolution blocks and we test the across-architecture performance on ResNet18, ResNet50, and ResNet101.
>
>      |              | ResNet18 | ResNet50 | ResNet101 |
>      | :----------: | :------: | :------: | :-------: |
>      |     Ours     |  57.97   |  31.29   |   25.66   |
>      |   Baseline   |  45.05   |  22.19   |   16.74   |
>      | Real Dataset |  91.74   |  90.74   |   90.01   |
>
>      We find that the gap of performance with the compressed dataset and the real dataset is increasing with the growth of model size. We think that it would become tough for the synthetic dataset with such a high compression ratio to train up a huge model with significantly more parameters, which is a challenge for dataset distillation.

---

> > ### Comment · Reviewer_jG3P · 2022-08-09
> > **Thank you for the detailed response**
> >
> > Many thanks to the author's responses. My concerns have been addressed.

---

### Official Review · Reviewer_fSqe · 2022-07-11

**Rating:** 5
**Confidence:** 4
**Soundness:** 3 good
**Presentation:** 3 good
**Contribution:** 2 fair

**Summary:**

The paper investigates different ways to parametrize the synthetic learned/distilled dataset in the Dataset Distillation [1]. In particular, they propose to reparametrize the dataset as $\{G_j(z_i)\}_{ij}$, where $z_i$'s are "basis" (i.e., latents in usual generative modelling terminology), and $G_i$'s are the "hallucinators" (i.e., generators). (I took liberty to use variable $z$ rather than the $\hat{x}$ in paper, for better clarity).

This reparametrization can be essentially applied with any Dataset Distillation method. In particular, the authors show that combining it with MTT (the SOTA Datasett Distillation work), can learn better distilled datasets **with same total number of parameters but with a much larger dataset size**. In fact, the performance gain mostly come from the increased dataset size (see below).


[1] Tongzhou Wang, Jun-Yan Zhu, Antonio Torralba, and Alexei A Efros. Dataset distillation. arXiv preprint arXiv:1811.10959, 2018.

**Questions:**

1. **Why should we care about lower parameter count? Comparison with standard image compression techniques?**

   Dataset Distillation (DD)'s scientific inquiry and practical applications both rely on the smaller distilled dataset size. Are there any applications that strongly relies on parameter counts? Indeed, smaller storage cost is always better, but I'm not sure if it matters that much when the distilled dataset often just contain tens or hundreds of images. Suppose that we care, then the authors should compare with regular compression techniques, including lossless and lossy ones, image-specific and generic ones.

   As an example, a simple lossless GIF compression of [a publicly available image from the distilled dataset of CIFAR-10 trained using MTT](https://georgecazenavette.github.io/mtt-distillation/images/cifar10_zca_10/airplane_4.png) is just 1923 bytes (with meta info!), versus the raw 3072 bytes.  It should be easy to compress much further, and essentially get much smaller parameter count.

2. **Given the importance of distilled dataset size in Dataset Distillation research, the paper should also compare methods at the same dataset size.**

   E.g., the proposed method at **BPC(bases-per-class)=10** has **IPC(images-per-class)=50**, but is only compared with other methods run at **IPC=11**. In fact, when fixing the dataset size, the proposed method is often slightly inferior (e.g., BPC=10 vs IPC=50).

   The current comparison is done at the same parameter count, which is only meaningful if the emphasis on parameter count cannot be sufficiently justified (1st question). In either case, same dataset size (IPC) comparison should be done.

3. **The stated motivation is not justified or verified.**

   The motivation for the proposed bias-hallucinator parameterization  is that the original pixel parametrization don't learn relations among samples. (lines 29-33)

   1. What exactly is "relations" referring to here?

   2. The pixel parametrization is the most flexible and can represent any images. Admittedly, different parametrizations lead to different inductive biases / implicit regularizations. Is it correct to understand the claim as a hypothesis that the bias-hallucinator parameterization has better inductive bias to learn better distilled datasets due to sharing the same hallucinator across biases?

   If so, the hypothesis seems false from the experiment results (see 2nd question above).  In other words, the stated performance gain mostly come from the increased distilled dataset size, rather than from the proposed parametrization.

3. **Motivating example in Section 1 is misleading.**

   It is a comparison among **(1)** proposed parameterization **(2)** original pixel parametrization w/ same #parameters and thus much less \#images **(3)** same as (2), but combining images from different checkpoints to form a larger distilled dataset.

   (3) is a really bizarre and misleading choice because the original parametrization is perfectly capable of learning a larger distilled dataset, and (3) is not really reducing any \#parameter or storage cost either. In fact, the later experiment results show that simply using original parametrization to distill a larger set have similar performance as (1). This makes the choice of (3) quite suspicious.

3. **This is essentially generative modelling with a discrete set of latents. Why use image-shaped "basis" rather than generic vectors?**

   It is not really wrong, but a somewhat weird choice given the common practice of using generic latent vectors. An explanation would be better to have.

4. **Please emphasize somewhere early in the paper that the increased dataset size is needed to achieve good performance.**

   A reader should not have to reach experiment section and to do the arithmetic to realize that the proposed technique comes at a cost.

5. **Please reconsider the choice of using DC to refer to the task and writing the paper as if DC introduced the task (see first section).**

6. The paper have some quite confusing places. Here are a few:


     1. Fig. 3 mentions a $\mathcal{L}_{cts.}$ that is not mentioned anywhere else.

     3.  " Constrain" => " Constraint" in multiple places.

**Limitations:**

I do not think that the limitation is sufficiently discussed. As I emphasized above, the authors should make it clear that the claimed improvements come with the cost of much larger distilled dataset size, the one metric that Dataset Distillation research cares about the most.  Furthermore, the added generator components should increase both training time and the training GPU memory cost. The training time effect is not really discussed other than a sentence in the checklist, which is not part of the paper.

**Strengths And Weaknesses:**

I will first discuss what I believe are two important issues of the current paper, and then provide a list of strengths and weaknesses.

### **Parameter count and dataset size**

It is important to note that
+ The method essentially is a **compression** of the distilled dataset in terms of #parameters, but **not in** dataset size, the usual quantity of interest in Dataset Distillation. The authors unfortunately do not make this clear in most cases when claiming improvements, which in reality comes at a cost of larger distilled dataset size.
+ It is unclear why #parameters should be a meaningful metric for Dataset Distillation. The paper does not provide arguments for it, or compare with any image compression methods.
+ When comparing with same **distilled dataset size**, the proposed approach often obtains slightly inferior performance (than the original pixel parameterization). This runs directly in contrary of the authors' claim that optimizing pixels directly is difficult for learning relations between samples (lines 29-33,43-44,121-124,329-331).

### **Improper credit assignment in writing**

I believe that it is greatly damages the integrity and openness of the academic community to
+ Refer to this task as Dataset Condensation (DC) [2] rather than Dataset Distillation (DD) [1]
+ Write as if the DC paper [2] introduces this task (entire paper, esp. introduction and abstract; DD is only mentioned once in related work and as a row in results table),

when
+ Both DD [1] and DC [2] talk about the same task, with DD paper predating DC paper for **2 years**, DC paper citing DD paper and acknowledging that they investigate the same task,
+ Neither DD paper nor DC paper explicitly gives a name to the task, but calls their proposed method DD or DC, and seems to refer to the task as DD or DC.

It is **extremely misleading** to people not familiar with this area, and essentially assigns credit in an obviously wrong way. I sincerely hope the authors did not do this purposefully. Admittedly, some other papers do the same thing, but it is no excuse to keep doing the harm. I strongly urge the authors to revise the paper in this aspect.

### **List of Strengths and Weaknesses**

**Strengths:**
+ The authors show that, by reparametrization with deep generators in Dataset Distillation training, it is possible to further compress a distilled dataset into fewer parameters, without losing much performance.
+ Therefore, with the same parameter count, the parametrization achieves better Dataset Distillation performance (at the cost of larger distilled datasets). If parameter count turns out a useful metric in future, this can potentially have better use cases, after validating its benefits against other image compression techniques.

**Weaknesses:**
+ ~The stated motivation for proposed reparametrization is that (original) pixel parametrization don't learn relations. This claim is not immediately clear why it is true, and is not verified. In fact, results in paper show that parametrization does not help when the dataset size is kept the same.~
+ ~The motivating example in Section 1 is misleading.~
+ The paper mentions reduced storage cost, but it is unclear why it should matter when the distilled dataset often only contains tens or hundreds of images.
+ ~Writing is somewhat misleading in that whenever performance improvement is mentioned, the increased dataset size is almost never mentioned.~
+ ~No comparison with standard image compression techniques.~
+ Missing comparison with other DD parametrization work.

Please see my questions and suggestions below.

[1] Tongzhou Wang, Jun-Yan Zhu, Antonio Torralba, and Alexei A Efros. Dataset distillation. arXiv preprint arXiv:1811.10959, 2018.

[2] Bo Zhao, Konda Reddy Mopuri, and Hakan Bilen. Dataset condensation with gradient matching. arXiv preprint arXiv:2006.05929, 2020.

----
The author's comment addressed most of my concerns (striked through text above). I have adjusted my score accordingly.

---

> ### Author Response · Authors · 2022-08-02
> **Response to Reviewer fSqe (Part IV)**
>
> 5. **Why use image-shaped "basis" rather than generic vectors?**
>
>    * Thanks for the pertinent comment. We adopt this setting mainly for better explainablity.
>
>    * In our conception, the bases are expected to capture basic information of images such as semantics and contours of contents, while hallucinators are expected to render the appearances such as colors and styles, which enjoys better explainablity as those shown in Fig. 4 and supplementary materials. Under this design, we consider the widely used encoder-decoder framework for pixel-wise image translation. Besides, since our bases share the same shape, or parameterization, with the raw images, it is more convenient and remarkable for this setting to reflect the advantage of our framework over the baselines.
>
>    * The reviewer’s proposal to use generic vectors is definitely feasible and promising. In this setting, the encoder of the hallucinators can be removed. We provide the following experimental accuracy (%) on the CIFAR-10 dataset for different ways of parameterization. All the comparisons are conducted with the number of total parameters held the same.
>      |       BPC        |       1        |       10       |       50       |
>      | :--------------: | :------------: | :------------: | :------------: |
>      |       Ours       | 55.66$\pm$0.29 | 70.27$\pm$0.63 | 74.04$\pm$0.16 |
>      | Ours w/o Encoder | 53.89$\pm$0.20 | 69.66$\pm$0.54 | 73.91$\pm$0.26 |
>      |     Baseline     | 49.89$\pm$0.95 | 65.92$\pm$0.62 | 70.73$\pm$0.52 |
>
>      We can find that the experimental results for the two settings are similar.
>
> 6. **Please emphasize somewhere early in the paper that the increased dataset size is needed to achieve good performance.**
>
>    * Sorry for the inconvenience. As explained above, we do not increase the dataset size. It is true that the number of final images increases without introducing additional cost of storage, which has been indicated in Line 56 to 61 of the revised version.
>
> 7. **Please reconsider the choice of using DC to refer to the task and writing the paper as if DC introduced the task.**
>
>    * Thanks for the comment. We have submitted the Rebuttal Revision which addresses this issue.
>
> 8. **Some typos.**
>
>    * Thanks for these detailed inspections. $L_{cts.}$ should be $L_{con.}$ and "Constrain" should be "Constraint". We have fixed them in the Rebuttal Revision.
>
> 9. **Limitations.**
>
>    * Thanks for pointing out some potential limitations. We admit that the added generator components makes inferior training efficiency of dataset distillation compared with baselines, including time and GPU memory cost, as indicated in Line 223 to 224. That is why we hope to enhance the training efficiency of the proposed method in future works, as indicated in Line 343. We have made a revision to add these potential limitations in the supplementary material of the revised version.
>
> ***
>
> [1] Dataset Distillation, Wang et al., arXiv, 2018
>
> [2] Dataset Condensation with Gradient Matching, Zhao et al., ICLR, 2021
>
> [3] Dataset Condensation with Differentiable Siamese Augmentation, Zhao et al., ICML, 2021
>
> [4] Dataset Condensation with Distribution Matching, Zhao et al., arXiv, 2021
>
> [5] Cafe: Learning to Condense Dataset by Aligning Features, Wang et al., CVPR, 2022
>
> [6] Dataset Distillation by Matching Training Trajectories, Cazenavette et al., CVPR, 2022
>
> [7] Dataset Condensation via Efficient Synthetic-Data Parameterization, Kim et al., ICML 2022
>
> [8] Remember the Past: Distilling Datasets into Addressable Memories for Neural Networks, Deng et al., arXiv, 2022

---

> ### Author Response · Authors · 2022-08-02
> **Response to Reviewer fSqe (Part III)**
>
> 3. **The stated motivation is not justified or verified.**
>
>    * The motivation of our method is to store some common knowledge shared by samples in a dataset in hallucination networks. In this sense, the networks should be capable of extracting some sample-wise relations. For example, when different bases are sent to the same hallucinator, the images decoded by this hallucinator should demonstrate some shared property, which can be viewed as relations. By contrast, in the traditional parameterization, each image has to store a copy of such common knowledge. Thus, it does not learn relations among samples.
>    * We agree with the reviewer that our hallucinators can capture better inductive bias. With such common inductive bias  / knowledge encoded, our method provides the bases with more freedom to learn other more useful and sample-specific knowledge. In this way, it is capable of learning information of $|\mathcal{H}|\times|\mathcal{B}|$ images with $|\mathcal{H}|$ hallucinators and $|\mathcal{B}|$ bases.
>    * In other words, the proposed parameterization enables the increased number of images with comparable or even less storage and downstream training burden, which is a free lunch to incorporate more images.
>
> 4. **Motivating example in Section 1 is misleading.**
>
>    * Here, this is a motivating example to demonstrate that the original parameterization can even be improved by a simple trick, *i.e.*, combining images from different checkpoints. The impact of this behavior is introducing some closely related but different samples, which increases the diversity. However, such a naive way introduces additional cost of storage. Thus, we consider encoded the shared relationship with shared hallucinator networks, to achieve similar effect without increasing the cost of storage.
>    * The main purpose of this example is not to compare with the baseline method under the same number of images. Thus, we do not choose to use the baseline method to learn more images directly. This evaluation can be found above and has been added to the revised version.

---

> > ### Comment · Reviewer_fSqe · 2022-08-07
> > **Reply to authors**
> >
> > Thanks for the response! I really appreciate the updates (esp. the credit assignment changes), answers and new results! It cleared up most of my confusions and I have raised my score accordingly. I have a few more questions:
> >
> > 1. [1] can be also viewed as a DD compression work, where a distilled dataset is compressed to be parametrized with smaller parameter counts. ~Would the authors consider comparing with them?~
> >
> >
> >   EDIT: after realizing that code of [1] is only released a few days before NeurIPS deadline, I don't think that it was reasonable that I asked for a comparison. However, I am curious to hear the authors' thoughts on how works like [1] are related to the present paper.
> >
> > 2. Reducing distilled dataset #parameters can be very important in continual learning applications. Prior works (e.g., [2,3]) show that storing distilled dataset for each task can help reduce catastrophic forgetting. Here the actual storage size is important and the proposed HaBa can potentially be very useful. Would the authors consider doing such experiments? I would imagine that HaBa can lead to important gains, which can better justify the focus on storage size (#parameter).
> >
> > [1] Jang-Hyun Kim, Jinuk Kim, Seong Joon Oh, Sangdoo Yun, Hwanjun Song, Joonhyun Jeong, Jung-Woo Ha, and Hyun Oh Song. Dataset  condensation via efficient synthetic-data parameterization. arXiv preprint arXiv:2205.14959, 2022.
> >
> > [2] Lee, Saehyung, Sanghyuk Chun, Sangwon Jung, Sangdoo Yun, and Sungroh Yoon. "Dataset Condensation with Contrastive Signals." arXiv preprint arXiv:2202.02916 (2022).
> >
> > [3] Zhao, Bo, and Hakan Bilen. "Dataset condensation with differentiable siamese augmentation." In International Conference on Machine Learning, pp. 12674-12685. PMLR, 2021.

---

> > > ### Author Response · Authors · 2022-08-09
> > > **Response to Reviewer fSqe**
> > >
> > > We sincerely thank the reviewer fSqe for the reply and are very happy to see that most concerns have been addressed. Here we would also like to have following discussions on the mentioned questions:
> > >
> > > 1. **How is the IDC work [1] related to the present paper?**
> > >
> > >    * Thanks for bringing a concurrent work to our attention. Indeed, the paper of IDC was submitted to arXiv after the submission deadline of NeurIPS 2022. Nevertheless, we would love to provide a discussion and comparison here and in the revision.
> > >
> > >    * For the contribution on the parameterization part in IDC, the work [1] reveals that only storing down-sample version of synthetic images and conducting bilinear upsampling in downstream training would not hurt the performance much. Thus, given the same budget of storage, it can store $4\times$ number of $2\times$ down-sample synthetic images compared with the baseline.
> > >
> > >    * According to the definition of our hallucinator-basis factorization, IDC can in fact be treated as a special case of HaBa, where the hallucinator is a parameter-free upsampling function and each basis has a smaller spatial size.
> > >
> > >    * The main focuses for IDC and HaBa are however different. For IDC, the core is to reduce the spatial size for efficient parameterization. For HaBa of this paper, instead, we do not modify the spatial size of bases in the default setting for better explainablity and intuitive comparisons with the baselines.
> > >
> > >    * Thus, IDC and HaBa are in fact two orthogonal techniques and they can readily join force to enhance the baseline performance. Here, we try using the technique of IDC and adopting $2\times$ down-sample synthetic images on the baseline MTT, based on which we further consider adding our HaBa and involving 5 hallucinators. The results are as follows:
> > >
> > >      | Avg. # of Param / Class | 2$\times$32$\times$32$\times$3 | 11$\times$32$\times$32$\times$3 | 51$\times$32$\times$32$\times$3 |
> > >      | :---------------------: | :----------------------------: | :-----------------------------: | :-----------------------------: |
> > >      |        Baseline         |         49.89$\pm$0.95         |         65.92$\pm$0.62          |         70.73$\pm$0.52          |
> > >      |      Baseline+IDC       |         56.13$\pm$0.38         |         70.85$\pm$0.43          |         71.01$\pm$0.41          |
> > >      |    Baseline+IDC+ours    |         61.27$\pm$0.34         |         72.14$\pm$0.22          |         75.31$\pm$0.27          |
> > >
> > >      We can observe that with the efficient parameterization of IDC, the performance of baseline can be improved. With HaBa in this paper, the performance can be further improved a lot: 5.14%, 1.29%, and 4.30% in the three settings respectively, which demonstrates that IDC and HaBa work in different ways.
> > >
> > > 2. **Applications in continual learning.**
> > >
> > >    * We appreciate the reviewer for pointing out the setting of continual learning (CL), an application that can potentially demonstrate the advantage of the proposed solution further.
> > >
> > >    * Following the setting of DM [2], we conduct experiments on the CL setting of CIFAR-100, with 20 random classes per stage. The average number of parameters per class is $20\times32\times32\times3$. The synthetic datasets are trained using ConvNet. The accuracy results for our method and the DM baseline on the same ConvNet architecture are as follows:
> > >
> > >      | # of Class |  20  |  40  |  60  |  80  | 100  |
> > >      | :--------: | :--: | :--: | :--: | :--: | :--: |
> > >      |  Baseline  | 57.3 | 50.8 | 46.5 | 42.3 | 38.8 |
> > >      |    Ours    | 63.4 | 56.9 | 52.0 | 47.2 | 43.7 |
> > >      |    Gain    | +6.1 | +6.1 | +5.5 | +4.9 | +4.9 |
> > >
> > >      On ResNet-18 architecture, the results are as follows:
> > >
> > >      | # of Class |  20   |  40   |  60   |  80   |  100  |
> > >      | :--------: | :---: | :---: | :---: | :---: | :---: |
> > >      |  Baseline  | 48.7  | 39.9  | 34.7  | 30.8  | 27.2  |
> > >      |    Ours    | 59.6  | 52.6  | 47.0  | 42.4  | 38.6  |
> > >      |    Gain    | +10.9 | +12.7 | +12.3 | +11.6 | +11.4 |
> > >
> > >      Above results demonstrate that the proposed method can indeed increase the informativeness of synthetic datasets and thus produce significantly better performance, especially on the cross-architecture setting.
> > >
> > > We have included these discussions and results in the revised supplement. Thanks again for the constructive suggestions. Sincerely hope that our response clarifies the reviewer's questions.
> > >
> > > ***
> > >
> > > [1] Dataset Condensation via Efficient Synthetic-Data Parameterization, Kim et al., ICML 2022
> > >
> > > [2] Dataset Condensation with Distribution Matching, Zhao et al., arXiv, 2021

---

> > > > ### Comment · Reviewer_fSqe · 2022-08-09
> > > > **Reply to authors**
> > > >
> > > > Thanks a lot for the update and running the experiments! I'm really impressed by how fast the results are produced :) They also further prove the usefulness of HaBa (esp. CL).
> > > >
> > > > In the original draft, the motivation for HaBa was not really convincing to me. Part of the paper (1st half) argues about some shared knowledge among samples. If the experiments are to support this claim, then they should really highlight how the parametrization is helpful when total #images is held constant. Yet 2nd half of the paper (experiments) focuses on how results are improved with similar #parameter, which would be fine if the goal is to reduce #parameters (after justifying this goal).
> > > >
> > > > This mismatch led to my confusion on the motivation and importance of the paper. I would sincerely appreciate if the authors could think more about how to better present the work in a more consistent manner.
> > > >
> > > > That said, the authors' replies and results have clarified my confusion in the rebuttal period. I now better understand the motivation (sharing knowledge) and am happy to see that the "side effect" (lower #parameter/image) also improve CL. Thus, I have raised my score further.

---

> ### Author Response · Authors · 2022-08-02
> **Response to Reviewer fSqe (Part II)**
>
> If we can reach an agreement on the metric, we think most concerns could be addressed:
>
> 1. **Why should we care about lower parameter count? Comparison with standard image compression techniques?**
>
>    * Actually, the main purpose of this paper is not to further decrease the parameter count, which is indeed already very low as discussed. Instead, we would like to increase the average information contained by each parameter. Thus, in all the experiments, the parameter counts of our method and baseline methods are held the same. The results indeed reflect some advantages of our method.
>
>    * Thanks for pointing out the image compression, a closely related research area, to us. In fact, dataset distillation and image compression are orthogonal, which means they can be adopted together to decrease the cost of storage, for both our method and baselines. As bases in this paper have the same shape / format with original images, image compression techniques are also applicable to our method. If we take the jpeg image compression technique into consideration, we have the following results on CIFAR-10 dataset with 10 bases per class for our method and 11 images per class for the baseline:
>
>      |                  |  Accuracy (%)  | # of Parameters | # of KB after Compression |
>      | :--------------: | :------------: | :-------------: | :-----------------------: |
>      |     Baseline     | 65.92$\pm$0.62 |     337,920     |           112.2           |
>      |       Ours       | 70.27$\pm$0.63 |     335,040     |           120.1           |
>      | Ours - 1 channel | 68.98$\pm$0.44 |     129,970     |           96.3            |
>
>      We observe that image compression technique can have a comparable contribution to the cost of storage for both our method and the baseline, which means it is indeed orthogonal with dataset distillation in this paper. Moreover, we also find that we can even use 1-channel bases to reduce the cost of storage further without hurting much performance.
>
> 2. **Comparison at the same dataset size.**
>
>    * We agree with the reviewer that comparing at the same number of images is also meaningful in dataset distillation to reflect some properties. In this paper, if our method and the baseline use the exact same objective function for dataset distillation, the upper bound of information carried by our parameterization, which contains much less parameters, cannot exceed that by the original one, since the baseline method directly conducts optimization over the resulting images. As such, performance of the baseline, in reality, imposes  **an upper bound** of that of our method when the same objective function is adopted, *i.e.*, w/o $\mathcal{L}_{cos.}$. We show the comparison on CIFAR-10 dataset as follows:
>
>      |         \# of Images          |      10      |      20      |      30      |      40      |      50      |
>      | :---------------------------: | :----------: | :----------: | :----------: | :----------: | :----------: |
>      |           Baseline            | 65.3$\pm$0.7 | 68.6$\pm$0.5 | 70.4$\pm$0.5 | 71.2$\pm$0.3 | 71.6$\pm$0.2 |
>      |             Ours              | 65.3$\pm$0.3 | 68.9$\pm$0.3 | 70.8$\pm$0.4 | 71.8$\pm$0.3 | 72.2$\pm$0.3 |
>      | Ours w/o $\mathcal{L}_{cos.}$ | 64.4$\pm$0.4 | 67.7$\pm$0.4 | 69.4$\pm$0.6 | 70.8$\pm$0.3 | 71.6$\pm$0.2 |
>
>    * Here for our method, we use 2 hallucinators and the number of bases is the number of images / 2. We can observe that under this setting, our performance can indeed approximate the theoretical upper bound of the baseline, with much less cost of storage and downstream training time.
>
>    * With the proposed adversary contrastive constraint, however, our method can even outperform the baseline consistently, which further demonstrates the effectiveness of the proposed solution.
>
>    * We have added the discussions and experiments on this aspect in Fig. 6 and the corresponding texts of the revised version.

---

> ### Author Response · Authors · 2022-08-02
> **Response to Reviewer fSqe (Part I)**
>
> We sincerely thank the reviewer fSqe for the pertinent feedback and are happy that the reviewer finds our method achieves better dataset distillation performance in terms of the same parameter count. There are two major concerns:
>
> 1. Using parameter count or the number of final images as the evaluation metric;
> 2. Referring to the task as Dataset Distillation (DD) [1] or Dataset Condensation (DC) [2].
>
> We would like to address the 2nd concern firstly since it is not related to the technical part. We agree that the two names refer to the same task and the concept of the task is introduced by the DD paper initially. We apologize for the improper credit assignment, and we did not intend to do so but merely followed the naming convention of previous works. We have now revised the paper, and have changed our title to “Dataset Distillation via Factorziation”, to highlight the credit of DD. Moreover, we have revised various parts of the manuscript,  including abstract, introduction, and related works as suggested, to clarify the relationship between DD and DC, and to acknowledge the pioneering contribution of DD. Please refer to the Rebuttal Revision. We sincerely appreciate the reviewer bringing this to our attention. We hope that in this version the relationship between DD and DC is clearly clarified.
>
> For the 1st concern, our opinion is that **parameter count is a reasonable metric for the area of dataset distillation under the setting of comparison in this paper**. We have the following reasons:
>
> * The motivations of dataset distillation are two-fold: to **alleviate the burden on storage** [2] and **speed up the training of downstream models** [1]. Let us discuss them respectively:
>   1. The cost of storage can be reflected directly by the number of parameters. What users actually care about is how to use lowest storage to obtain the most information, instead of how many images the dataset has.
>   2. The proposed method does not increase the time cost of training downstream models. Actually, in all the comparisons, the dataset size is set as the number of bases while hallucinators are treated as a kind of parameterized data augmentation, which works online in downstream training just as general data augmentation techniques. In this sense, **the actual dataset size for one epoch does not increase**. In other words, we rigorously control the training **iterations** and **batch size**, used by our method and baseline methods the same. Moreover, since the hallucination networks used in this paper are lightweight, the time cost of online image composition is actually close to general data augmentation techniques (**140.11 v.s. 142.86 epochs per second** on CIFAR-10 with 10 bases / images per class). These facts indicate that our method can improve the performance significantly given less cost of storage and almost the same downstream training time. We have clarified this in Line 226 to 229 of the revised version.
> * Using the number of final images as the evaluation metric, unfortunately, indeed suffers from significant drawbacks:
>   1. If we consider data augmentation, one of the simplest techniques, horizontal flip, can make the number of images double. If other techniques like random rotation, random shift, and color jitter, the number of images becomes infinite. In this sense, the size of distilled datasets in almost all the previous methods [3,4,5,6] on dataset distillation are infinite, which becomes weird.
>   2. There may be many trivial solutions if only the number of images is used as the evaluation metric. For example, we can concatenate all the images in a dataset to a single one to form a huge image. In this case, we get a dataset whose size is only 1.
>
> Thus, we believe that parameter count is truly reasonable to be adopted as the main comparison metric, which is also the common comparison scheme in recent works focusing on the parameterization of the distilled data [7,8].

---

### Official Review · Reviewer_9m6U · 2022-07-11

**Rating:** 7
**Confidence:** 5
**Soundness:** 3 good
**Presentation:** 3 good
**Contribution:** 3 good

**Summary:**

This paper proposes a new parameterization (HaBa) of image data for dataset condensation. The proposed method decomposes a dataset into two components: data hallucination networks and bases, and considers the relationship between different condensed data points. Experiments show that HaBa achieves a better compression rate and better cross-architecture generalization performance.

**Questions:**

3.1 Basis and Hallucinator:
- I am curious about the functionality of the encoder of the Hallucinators. Is it necessary? What will happen if you remove the encoder part and only train the decoder and corresponding style vector?
- I wonder if it would be more parameter-efficient to learn multiple style vector pairs ( $\sigma$ , $\mu$ ) for a particular encoder and decoder pair.
- I guess the weights can learn more information than the fixed basis. Why do you only use 1 Conv-ReLu block for Hallucinators? What will happen if you consider more layers in the encoder and decoder structure? Why not consider increasing the capacity of the hallucinator by expanding its width?

3.2 Adversarial Contrastive Constrain
- Why do you want a contrastive objective for the feature extractor and a cosine similarity for the hallucinator? Could you have just one objective (e.g., cosine objective) for both the feature extractor and the hallucinator (the feature extractor maximizes it, and the hallucinator minimizes it)?

4.1 Datasets and Implementing Details
- How long does the training take with the new parameterization compared to baselines? Is the optimization easier or harder with the new parameterization?
- I can imagine that it is possible that some combination of hallucinator and basis may not be appropriate and introduce bias into the downstream classifier. For example, the dog's style may not be suitable for a ship. Suppose you train the downstream classifier on a ship instance with dog style. Would the classifier become confused at test time? What is your opinion on that?
- Because the proposed methods have so many hyperparameters, I suggest adding a table summary of the hyperparameters used for each experiment in the appendix.

4.3 Ablation Studies
- Cross-Architecture Performance: Lack of explanation. Why does it improve the cross-architecture performance? What is the insight behind it? What design choice gives this nice property?
- Cross-Domain Performance: Maybe change the title to "robustness to corruption"? I am unclear why "diverse training data" can give this nice property.
- I would like to see some ablation studies on what will happen if we increase the hallucinator capacity (increase depth or width).

Misc
- Line 63: "Such an adversarial scheme, in turn, enforces the hallucinators to produce diversified images and recover original data distribution thoroughly." The argument that "recover original data distribution" appears many times in the paper. I don't think dataset condensation methods aim to recover the data distribution. They are mining for informative examples instead. How come the condensed data recovers the original data distribution? Visually, the condensed data looks very different from the real data. What is your opinion on this? I would be more comfortable with an argument like "capture the discriminative feature better."
- Line 89: "supervision signals from downstream tasks, e.g., cross-entropy loss for classification, are not rigorous enough for synthetic samples." What do you mean by "rigorous enough"? What do you consider a rigorous supervision signal?
- Line 211: Typo. 10, 100, and 200 -> 10, 100, and 100
- Figure 4 and Table 1 are far away from their text, which disrupts the reading flow.
- Table 4 is confusing since it includes two different comparisons. Splitting into two tables or moving one to the appendix seems better.

I would be happy to increase my score if my questions can be addressed appropriately.

**Limitations:**

The authors may want to discuss the limitations of their methods. Here are some of my incomplete thoughts.
- There are so many hyperparameters in the proposed method that may impede its practical usage.
- The parameterization only works for image data but may want to consider other modalities.
- When scaling the method to many bases and hallucinators, there may be some optimization issues.

**Strengths And Weaknesses:**

This paper is well-motivated and focuses on an essential problem (the parameterization of the condensed data) for current dataset condensation methods. The proposed method is novel, and the results are encouraging.

- Originality: Decomposing the condensed data to bases and hallucinators is novel and interesting. Related work focusing on the parameterization of the condensed data: https://arxiv.org/abs/2205.14959, https://arxiv.org/abs/2206.02916.
- Quality: Most arguments are well-supported, and the experiments demonstrate the effectiveness of the proposed method. Some additional ablation studies and experiments are needed to make the paper more convincing.
- Clarity: This paper is well-written and easy to follow, though the position of some figures and tables can disrupt the reading experience. Some training details can be added to the appendix.
- Significance: The experiments demonstrate the effectiveness of the proposed method and can inspire some future work on the parameterization of the condensed data.

---

> ### Author Response · Authors · 2022-08-02
> **Response to Reviewer 9m6U (Part IV)**
>
> ### About 4.3 - Ablation Studies
>
> 1. **Why does the proposed method improve the cross-architecture performance?**
>    * Thanks for the question. Actually, the proposed method brings a more informative parameterization of condensed datasets. With the scheme of factorization, information of $|\mathcal{H}|\times|\mathcal{B}|$ images can be included with $|\mathcal{H}|$ hallucinators and $|\mathcal{B}|$ bases. More useful amount of data would be beneficial for training downstream models in machine learning. Please refer to the TSNE visualization in the supplementary material.
> 2. **Why can diversifying training data increase the robustness to corruption?**
>    * Thanks for the question. In the theory of learning from different domains [1], the test error in the target domain is dominated by 1) the error in the source domain and  2) the distance between source and target domains. We would like to contribute the robustness to corruption to the former. As validated in the previous experiments, our method can achieve lower error on the original CIFAR-10 dataset. Thus, we may also expect better performance on the corresponding corrupted data compared with baseline methods.
>    * We have added more discussion in Fig. 5 and the corresponding texts of the supplementary material.
> 3. **Ablation studies on the depth and width of hallucinators.**
>    * Thanks for the suggestion. We have included these studies in the revised version. Please refer to **About 3.1 - Basis and Hallucinator - 3** for details of the experimental results.
>
> ### Misc
>
> * Thanks for these detailed inspections and terribly sorry for the unclarity. We have fixed all the issues in the revised version. Please refer to the Rebuttal Revision for details.
>
> ### Limitations
>
> 1. **About hyperparameters.**
>
>    * Thanks for the comment. Please refer to **About 4.1 - Datasets and Implementing Details - 3**.
>
> 2. **How does the proposed solution work for other modalities?**
>
>    * Thanks for pointing this out. In fact, the proposed dataset factorization can also work for other modalities. The principle is that bases can have the same parameterization with the raw data and hallucinators can be the translation model in the corresponding modalities.
>
>    * Here, we adopt the framework of IDC [7] as the baseline and conduct an experiment on Mini Speech Commands [8], a dataset for speech recognition, following their original setting. The results are as follows:
>
>      |   BPC    |  10  |  20  | Full Dataset |
>      | :------: | :--: | :--: | :----------: |
>      |   Ours   | 74.5 | 84.3 |     93.4     |
>      | Baseline | 73.3 | 83.0 |     93.4     |
>
>    * We have added these results in the revised supplementary material.
>
> 3. **Some optimization issues when scaling the method to many bases and hallucinators.**
>
>    * We agree with the reviewer that when the number of bases and hallucinators are large, there would be some optimization issues. As shown in Fig. 6 (right) of the main paper, further increasing the number of hallucinators does not result in substantial performance gain.
>    * Actually, such limitation is largely inherited from the adopted baseline methods. For example, when the number of images is large (greater than 50), the performance gain is also somewhat limited for MTT [6].
>
> We have added these potential limitations in the revised version.
>
> ***
>
> [1] A Theory of Learning from Different Domains, Ben-David et al., Machine Learning, 2010
>
> [2] Dataset Condensation with Distribution Matching, Zhao et al., arXiv, 2021
>
> [3] Cafe: Learning to Condense Dataset by Aligning Features, Wang et al., CVPR, 2022
>
> [4] Herding Dynamical Weights to Learn, Welling et al., Machine Learning, 2009
>
> [5] Active Learning for Convolutional Neural Networks: A Core-Set Approach, Sener et al., ICLR, 2018
>
> [6] Dataset Distillation by Matching Training Trajectories, Cazenavette et al., CVPR, 2022
>
> [7] Dataset Condensation via Efficient Synthetic-Data Parameterization, Kim et al., ICML 2022
>
> [8] A Dataset for Limited-Vocabulary Speech Recognition. Warden et al., arXiv 2018

---

> ### Author Response · Authors · 2022-08-02
> **Response to Reviewer 9m6U (Part III)**
>
> ### About 3.2 - Adversarial Contrastive Constraint
>
> 1. **Why do we use a contrastive objective for the feature extractor and a cosine similarity for the hallucinator?**
>
>    * Thanks for the insightful question. We empirically find that such configuration yields best results in practice. But it is also fine to consider other choices.
>
>    * The objective for the feature extractor is to minimize the divergence between two samples while one objective for our dataset factorization is to maximize such divergence. Intuitively, any meaningful metric, such as cosine similarity and mutual information, to reflect such divergence would be fine.
>
>    * In our early exploration, we indeed considered using the cosine similarity objective for both the feature extractor and the hallucinators. Later, we found that replacing it with mutual information for the feature extractor can produce slightly better performance, as shown in the results below. Since the feature extractor wants to minimize the divergence, or maximize the mutual information, we adopt the lower bound of the mutual information, which has the contrastive form.
>
>      |     BPC     |       1        |       10       |       50       |
>      | :---------: | :------------: | :------------: | :------------: |
>      |    Ours     | 55.66$\pm$0.29 | 70.27$\pm$0.63 | 74.04$\pm$0.16 |
>      | Ours (Cos.) | 55.61$\pm$0.94 | 70.09$\pm$0.39 | 73.34$\pm$0.36 |
>      |  Baseline   | 49.89$\pm$0.95 | 65.92$\pm$0.62 | 70.73$\pm$0.52 |
>
>    * If we analyze the form of the lower bound of the mutual information and the cosine similarity in detail, we can find that the difference is only on whether we should take negative samples into consideration, which can be viewed as a kind of normalization. Since directly enforcing a high similarity is probably harmful for the feature extractor to learn a discriminative representation, considering negative samples and regulating the relative similarity can perform better in this sense.
>
> ### About 4.1 - Datasets and Implementing Details
>
> 1. **How long does the training take with the new parameterization compared to baselines?**
>    * Thanks for the question. In the experiments, we use twice the number of iterations of that adopted in the official baseline setting.  We have to admit that the convergence indeed becomes slightly slower compared with that of the baseline method, which is an unfortunate fact given the nature of our task configuration.  In practice, however, this training efficiency is of minor concern for the dataset condesntaiton task, since the training take places for only once. As discussed in the conclusion and limitation, in our future work, we will explore improving the training efficiency of HaBa.
>
> 2. **Some combination of hallucinator and basis may not be appropriate.**
>    * Thanks for this pertinent point. Actually, in our exploration, we also experiment using an independent set of hallucinators for each class. As shown in Tab. 4 of the main paper, the performance gain is actually very limited, and it can increase additional cost on storage. That is why we adopt shared hallucinators across all the classes by default.
>    * Analyzed through the qualitative results in Fig. 4, we find it is not the case that one hallucinator can only generate one specific style. Instead, it can be somehow adaptive to the input bases. For example, if we input the bases of dogs to $H_1$, the generated images are equipped with reasonable styles of dog. And if we input the bases of ships to $H_1$, the styles are also reasonable to ships (the background is sea-like blue).
>    * We agree with the reviewer that it is promising to take class-wise relationships into consideration in future works. It is probably the optimal solution that one class shares hallucinators with some specific classes but does not share with others, which is more flexible. We have added it as a future direction in the revised supplementary material.
> 3. **About hyperparameters.**
>    * Thanks for the suggestion. Actually, most of the hyperparameters in HaBa come from the baseline method. As for the dataset factorization framework itself, the hyperparameters are only weights of those loss terms.
>    * We have added a summary of all the hyperparameters in Tab. 7 of the revised version as the reviewer suggested to provide a more convenient view.

---

> ### Author Response · Authors · 2022-08-02
> **Response to Reviewer 9m6U (Part II)**
>
> 3. **Why do we only use 1 Conv-ReLu block for hallucinators?**
>
>    * Thanks for the question. We use light-weight hallucinators with only 1 non-linear block, based on the trade-off between performance and data / training efficiency.
>
>    * Considering more hidden layers or expanding the width of hallucinators may potentially help learn more information, which can be validated by the following experiment on CIFAR-10 with 1 BPC:
>
>      |      \# of Non-Linear Block      |       0        |       1        |       2        |       3        |
>      | :------------------------------: | :------------: | :------------: | :------------: | :------------: |
>      |           Accuracy (%)           | 68.43$\pm$0.37 | 70.27$\pm$0.63 | 71.17$\pm$0.29 | 71.55$\pm$0.27 |
>      | Downstream Speed (epochs / sec.) |     144.54     |     140.11     |     125.04     |     115.62     |
>      |         \# of Parameters         |     6,144      |     6,312      |     10,963     |     16,131     |
>
>      |              Width               |       3        |       8        |       16       |
>      | :------------------------------: | :------------: | :------------: | :------------: |
>      |           Accuracy (%)           | 70.27$\pm$0.63 | 70.47$\pm$0.37 | 71.28$\pm$0.35 |
>      | Downstream Speed (epochs / sec.) |     140.11     |     138.48     |     135.12     |
>      |         \# of Parameters         |     6,312      |     16,827     |     33,651     |
>
>      The results show that increasing the depth from 1 to 2 makes the speed decrease by 10.8% but only has 0.9% performance gain. Increasing the width from 3 to 8 makes the number of parameters increased by 167% but only produces 0.2% performance gain.
>
>    * Although increasing depth or width of hallucinators can somehow increase the performance, we do not want to sacrifice too much on the cost of downstream training time and storage, both of which are important factors in dataset distillation. That is why we only consider light-weight hallucinators with 1 Conv-ReLu block, which yields the best trade-off between accuracy and efficiency.
>
>    * We have added discussions on this aspect in Tab. 3, 4 and the corresponding texts of the supplementary material.

---

> ### Author Response · Authors · 2022-08-02
> **Response to Reviewer 9m6U (Part I)**
>
> We sincerely thank the reviewer for the insightful and constructive comments. We are glad that the reviewer finds our work novel and interesting. The concerns are fully addressed as follows.
>
> ### About 3.1 - Basis and Hallucinator
>
> 1. **The functionality of the encoder of the hallucinators.**
>
>    * Thanks for the question. Since our bases have the same shape / format with final images, an encoder is required for feature extraction.
>
>    * In our conception, the bases are expected to capture basic information of images such as semantics and contours of contents, while hallucinators are expected to render the appearances such as colors and styles, which enjoys **better explainablity** as those shown in Fig. 4 and supplementary materials. Under this setting, we consider the widely used encoder-decoder framework for pixel-wise image translation. Since our bases share the same shape, or parameterization, with the raw images, it is more convenient and remarkable for this design to reflect what is the input of hallucination networks and better to understand the advantage of our framework over the baselines.
>
>    * The reviewer’s proposal to remove the encoder is definitely feasible and promising. It can be viewed as a variant of our formulation. In this setting, the bases become one type of latent code as those in typical generative models. We provide the following experimental accuracy (%) on the CIFAR-10 dataset for different ways of parameterization. All the comparisons are conducted with the number of total parameters held the same, which is also the setting for all following experiments.
>
>      |       BPC        |       1        |       1*       |       10       |      10*       |       50       |      50*       |
>      | :--------------: | :------------: | :------------: | :------------: | :------------: | :------------: | :------------: |
>      |       Ours       | 48.26$\pm$0.84 | 55.66$\pm$0.29 | 70.27$\pm$0.63 | 70.55$\pm$0.38 | 74.04$\pm$0.16 | 73.26$\pm$0.37 |
>      | Ours w/o Encoder | 47.71$\pm$0.77 | 53.89$\pm$0.20 | 69.66$\pm$0.54 | 68.98$\pm$0.44 | 73.91$\pm$0.26 | 71.52$\pm$0.23 |
>      |     Baseline     | 49.89$\pm$0.95 | 49.89$\pm$0.95 | 65.92$\pm$0.62 | 65.92$\pm$0.62 | 70.73$\pm$0.52 | 70.73$\pm$0.52 |
>
>      **Here, * denotes that we consider channel-independent basis, to send each channel of basis to hallucinators independently, which is equivalent to 1-channel basis. We can find that the experimental results of with encoder and w/o encoder are similar for 3-channel basis. But the performance drops a lot (greater than 1% accuracy) for 1-channel basis. We conjecture that 1-channel basis would more rely on encoder for sufficient feature extraction.**
>
> 2. **Using shared encoder and decoder across all the hallucinators.**
>
>    * Thanks for the advice. It can indeed be more parameter-efficient to learn multiple style vector pairs $(\sigma,\mu)$ for a particular encoder and decoder pair. It also can be viewed as a variant of our formulation. To reflect the impact of such design, we conduct an experiment which tends to use shared encoder and decoder across all the hallucinators:
>
>      |            BPC             |       1        |       1*       |       10       |      10*       |       50       |      50*       |
>      | :------------------------: | :------------: | :------------: | :------------: | :------------: | :------------: | :------------: |
>      |            Ours            | 48.26$\pm$0.84 | 55.66$\pm$0.29 | 70.27$\pm$0.63 | 70.55$\pm$0.38 | 74.04$\pm$0.16 | 73.26$\pm$0.37 |
>      | Ours w. Shared Enc. & Dec. | 46.47$\pm$0.19 | 55.14$\pm$0.44 | 69.47$\pm$0.09 | 70.22$\pm$0.37 | 72.69$\pm$0.39 | 72.99$\pm$0.04 |
>      |          Baseline          | 49.89$\pm$0.95 | 49.89$\pm$0.95 | 65.92$\pm$0.62 | 65.92$\pm$0.62 | 70.73$\pm$0.52 | 70.73$\pm$0.52 |
>
>      We can find that using a shared encoder and decoder may lead to a little bit inferior performance. We conjecture that different convolution encoders and decoders may contribute to the diversity of the extracted patterns, which increases the representation ability of the hallucinator set. Moreover, since we only use 1 convolution block for encoders and decoders, the number of parameters for encoder and decoder in a hallucinator (168) is not so significant compared with that of a basis (3072).
>
>    * We have added discussions on this aspect in Tab. 5 and the corresponding texts of the supplementary material.

---

> > ### Comment · Reviewer_9m6U · 2022-08-04
> > **Needs clarification**
> >
> > What is "baseline"? Is it MTT? Why does "Ours" achieve much better performance than the result in Table 1 for one image per class setting? What changes have been made?

---

> > > ### Author Response · Authors · 2022-08-06
> > > **Response to Reviewer 9m6U**
> > >
> > > We sincerely thank the reviewer for the further question. The baseline is MTT. The difference between the two results is on the **channel-independent basis** or **channel-dependent basis** adopted. In this paper, the shape of a basis is exact the same as that of an image. In other words, the number of channels in a basis is 3. Here, for the BPC=1 setting (IPC=BPC+1=2 for baseline), we consider channel-independent bases, to send each channel of bases to hallucinators independently, which is equivalent to 1-channel bases. In Tab. 1 of the main paper, to compare with previous baselines at the same setting, however, we do not adopt such manner and use 3-channel bases. In both cases and the baseline, the numbers of total parameters are the same. The difference between channel-independent bases and channel-dependent ones is analyzed in Fig. 7 of the main paper. We have also updated the aforementioned experimental results to provide the results for both channel-independent and channel-dependent settings. We hope our response clarifies the reviewer's question.

---

> > > > ### Comment · Reviewer_9m6U · 2022-08-07
> > > > **Response to Authors**
> > > >
> > > > Thanks for your clarification. I have another question regarding the scalability.
> > > >
> > > > I can see that many factors can affect the final performance, such as 1) width, depth, and the number of hallucinators, 2) size and the number of bases, 3) architecture of encoder and decoder, 4) the way to share a part of encoder or decoder. Besides, many trade-offs are going on among accuracy, speed, and the number of parameters. The authors have provided several ablation studies on each component independently, and all results agree with the expectation - "the larger, the better, and a diminishing return is going on." But it is unclear what is the best joint configuration given a fixed storage budget. I want to gain some insights on how the best configuration to achieve the best accuracy varies as we increase the storage budget. In other words, what is the best way to scale the bases and hallucinators? What are the dominant factors when the storage budget is small (large)? And when does the transition phase happen? To simplify the problem, let us ignore the speed and only focus on the accuracy and the number of parameters. How do we choose the width, depth, and the number of encoders (decoders)? How do we choose the number and the dimension of the bases? (I separate the encoder and decoder as you show that it can achieve good performance even without encoders). I think this problem is very important for practitioners if they want to use this new parameterization, and I feel like the current 1 Conv-ReLu block may not be the optimal choice.

---

> > > > > ### Author Response · Authors · 2022-08-09
> > > > > **Response to Reviewer 9m6U (Part II)**
> > > > >
> > > > > We provide **a more illustrating visualization** in the revised supplement and have the following observations:
> > > > >
> > > > > 1. **For all three types of budget, the best performance is achieved by using deeper hallucinators. Especially under small and medium budgets,  using depth 2 can outperform using depth 1 almost consistently.** This can be explained by the more complex sample-wise relationship extracted by hallucinators.
> > > > > 2. In our framework, bases are expected to store sample-independent information while hallucinators are used to encode shared relationships across all the samples. **When the budget is small, using 1-channel bases can achieve significantly better results.** This is because a small storage budget would rely more on increasing the number of independent data samples for better diversity. The informativeness of each basis appears less important.
> > > > > 3. **When the budget increases, the advantage of 1-channel bases mentioned before would diminish gradually. Especially under large budgets, 3-channel bases outperform 1-channel ones consistently.** The reason is that when the number of bases is adequate, focusing on the informativeness of each basis can produce more benefits than increasing the number.
> > > > > 4. **When the budget is large, using more hallucinators can yield slightly better results,** which can probably be attributed to the further improvement in the diversity.
> > > > > 5. **The larger the budget is, the less insensitive the performance is, to different configurations.**
> > > > >
> > > > > Note that the above exploration is conducted without taking the downstream training speed into consideration, which is also an important metric in the task of dataset distillation. Our opinion on the scalability is that, when downstream training overhead is not an issue, deeper hallucinators are recommended for better performance; otherwise if downstream efficiency is desired, we find that 1 nonlinear block is sufficient since heavier hallucination networks can result in nonnegligible latency, especially when the total number of images is large.
> > > > >
> > > > > We would like to thank the reviewer again, and we hope these discussions can bring some insights regarding the scalability.

---

> > > > > > ### Comment · Reviewer_9m6U · 2022-08-09
> > > > > > **Thanks for the response**
> > > > > >
> > > > > > Thanks for the response. I appreciate that the authors could conduct such an ablation study. It seems that in the small-budget region, the number of bases dominates the performance, while in the large-budget region, the expressivity of the hallucinators dominates the performance. If my understanding is correct, most parameters in the current HaBa are in the bases, which limits the expressive power. It seems to suggest that increasing the expressive power of the hallucinators can be a fruitful future direction when we want to distill tens of thousands of data. I would like to see the authors explore more in this direction in the future. I am satisfied with the author's efforts in the discussion phase so I will increase my score to 7 (Accept).

---

> > > > > ### Author Response · Authors · 2022-08-09
> > > > > **Response to Reviewer 9m6U (Part I)**
> > > > >
> > > > > We sincerely thank the reviewer for the further question on the  scalability. Indeed, under the framework of hallucinator-basis factorization, there are many factors that affect the performance. Given a fixed storage budget, how to scale the bases and hallucinators is an  important topic. Among all the factors, we empirically find that the  depth of hallucinators, the number of hallucinators, the number of  channels in each basis, and the number of bases are the most important  ones, which will be studied in the following exploration. Here, we consider three types of storage budget: small, medium, and large, corresponding to the cost of IPC=2, 11, and 51 for the baseline method respectively. We consider cases of 1 and 2 convolution blocks for the depth of hallucinators, 2 and 5 for the number of hallucinators, and 1 and 3 for the number of channels in each basis. For each setting, we adjust the number of bases to fit the given budget. The detailed configurations and results are as follows:
> > > > >
> > > > > | Storage Budget | Depth of Ha. | # of Ha. | # of Channels in Ba. | # of Ba. |      Accuracy      |
> > > > > | :------------: | :----------: | :------: | :------------------: | :------: | :----------------: |
> > > > > |     Small      |      1       |    2     |          1           |    5     |   56.57$\pm$0.10   |
> > > > > |     Small      |      1       |    2     |          3           |    2     |   51.91$\pm$0.73   |
> > > > > |     Small      |      1       |    5     |          1           |    3     |   55.66$\pm$0.29   |
> > > > > |     Small      |      1       |    5     |          3           |    1     |   48.26$\pm$0.84   |
> > > > > |     Small      |      2       |    2     |          1           |    4     | **62.02$\pm$0.35** |
> > > > > |     Small      |      2       |    2     |          3           |    1     |   58.44$\pm$0.18   |
> > > > > |     Small      |      2       |    5     |          1           |    1     |   60.96$\pm$0.33   |
> > > > > |     Medium     |      1       |    2     |          1           |    32    |   72.11$\pm$0.10   |
> > > > > |     Medium     |      1       |    2     |          3           |    11    |   69.02$\pm$0.30   |
> > > > > |     Medium     |      1       |    5     |          1           |    30    |   70.55$\pm$0.38   |
> > > > > |     Medium     |      1       |    5     |          3           |    10    |   70.27$\pm$0.63   |
> > > > > |     Medium     |      2       |    2     |          1           |    31    |   71.74$\pm$0.11   |
> > > > > |     Medium     |      2       |    2     |          3           |    10    | **73.76$\pm$0.10** |
> > > > > |     Medium     |      2       |    5     |          1           |    28    |   72.47$\pm$0.21   |
> > > > > |     Medium     |      2       |    5     |          3           |    9     |   71.76$\pm$0.22   |
> > > > > |     Large      |      1       |    2     |          1           |   152    |   73.07$\pm$0.20   |
> > > > > |     Large      |      1       |    2     |          3           |    51    |   74.59$\pm$0.32   |
> > > > > |     Large      |      1       |    5     |          1           |   150    |   73.26$\pm$0.37   |
> > > > > |     Large      |      1       |    5     |          3           |    50    |   74.04$\pm$0.16   |
> > > > > |     Large      |      2       |    2     |          1           |   151    |   73.06$\pm$0.23   |
> > > > > |     Large      |      2       |    2     |          3           |    50    |   73.26$\pm$0.42   |
> > > > > |     Large      |      2       |    5     |          1           |   148    |   73.76$\pm$0.27   |
> > > > > |     Large      |      2       |    5     |          3           |    49    | **75.44$\pm$0.22** |

---

### Meta-Review · Area_Chair_isWN · 2022-08-27

**Recommendation:** Accept
**Confidence:** Less certain

**Metareview:**

The reviewers originally had concerns but these have been well addressed by the authors in a thorough rebuttal and there is a consensus for acceptance. We encourage the authors to incorporate all the comments from the reviewers in the final version.

**Award:**

No

---

### Decision · Program_Chairs · 2022-09-14

Accept